# An Optimal Diffusion Approach to Quadratic Rate–Distortion Problems: New Solution and Approximation Methods

**Dror Freirich & Nir Weinberger** *
Viterbi Faculty of Electrical and Computer Engineering
Technion - Israel Institute of Technology
`drorfrc@gmail.com, nirwein@technion.ac.il`

## Abstract

When compressing continuous data, some loss of information is inevitable, and this incurred a distortion upon reconstruction. The Rate–Distortion (RD) function characterizes the minimum achievable rate for a code whose decoding permits a specified amount of distortion. We exploit the connection between rate-distortion theory and entropic optimal transport to propose a novel stochastic-control formulation for the former, and use a classic result dating back to Schrödinger to show that the tradeoff between rate and mean squared error distortion is equivalent to a tradeoff between control energy and the differential entropy of the terminal state, whose probability law defines the reconstruction distribution. For a special class of sources, we show that the optimal control law and the corresponding trajectory in the space of probability measures are obtained by solving a backward heat equation. In more general settings, our approach yields a numerical method that estimates the RD function using diffusion processes with a constant diffusion coefficient. We demonstrate the effectiveness of our method through several examples.

## 1 Introduction

As is well-known from information theory (Shannon, 1948; Cover and Thomas, 2012), when compressing any probabilistic data source at an encoding *rate* lower than its entropy, it is inevitable to suffer some loss of information, which results in a *distortion* at reconstruction. This is particularly true for *continuous* sources, where distortion may arise, *e.g.*, due to quantization. Rate distortion (RD) theory (Shannon, 1959; Berger, 2003; Cover and Thomas, 2012) addresses the trade-off between the encoding rate and the decoder's reconstruction: Given a distortion measure between pairs of data samples, the reconstruction loss is the total accumulated distortion between the source samples and reconstructed samples produced by the decoder from the compressed bits sent by the encoder. The *operational RD function* thus characterizes the minimal encoding rate required for a given average distortion level $D$, in the limit of a large number of samples.

For independent and identically distributed (*i.i.d.*) samples, the celebrated lossy source-coding theorem of information theory (Shannon, 1959; Berger, 2003; Cover and Thomas, 2012) establishes that the operational RD function equals the *informational RD function* $R(D)$, where the latter is expressed as a *single-letter expression* (though a single sample may itself be a vector). Specifically, the RD function is the minimum of the mutual information (MI) between a random variable $X$, representing a single sample from the source, and a random variable $\hat{X}$, representing a single reconstruction sample. This minimization is over the test channel – the conditional distribution of $\hat{X}$ given $X$ – under the constraint that the average distortion between $X$ and $\hat{X}$ is below $D$ (see § 2).

The optimization problem involved in the computation of $R(D)$ should therefore be solved for the given distortion measure and distortion level. However, although this problem has been extensively studied for almost 7-decades (Shannon, 1959), closed-form solutions of this problem are only known

---

*The authors wish to thank Lena Zugman and the Technion ECE library staff for their support.

for a few canonical examples, such as binary sources under Hamming distortion or a Gaussian source under the mean squared error (MSE) Cover and Thomas (2012). Efficient algorithms for computing the RD function are therefore necessary. When the data source alphabet is discrete and finite, the classical Blahut–Arimoto (BA) algorithm (Arimoto, 1972; Blahut, 1972) provides such a method and is especially effective for small alphabets. However, the typical setup for lossy compression involves a continuous data source, and in this setting efficient computation of the RD function remains a long-standing challenge. Recently, Lei et al. (2023); Yang et al. (2024) identified an interesting connection between the BA algorithm and entropy-regularized Optimal Transport (OT), which allows approximation of the RD function in cases where BA is intractable (Yang et al., 2024). OT (Ambrosio et al., 2008) is a widely used formulation, which underlies the training of deep generative models, such as generative adversarial networks (GANs) (Arjovsky et al., 2017) and variational auto-encoders (VAEs) (Tolstikhin et al., 2017), and is defined as follows: Given a pair of distributions, the objective of OT is to find an optimal plan between them, namely, a joint distribution with the given marginals (called a coupling) that minimizes a prescribed metric. *Entropy-regularized* optimal transport (*a.k.a. weak* OT, *entropic* OT, or *EOT*) was introduced by Cuturi (2013) as an approximation to OT, for which efficient solution methods exist, *e.g.*, Sinkhorn's algorithm (Altschuler et al., 2017). More recently, Gushchin et al. (2022) suggested finding the optimal plan of an EOT problem using diffusion processes. Their method is based on a well-known connection (Léonard, 2013; Chen et al., 2021) between EOT and a stochastic control problem known as the *Schrödinger bridge* (SB) (Schrödinger, 1932; Chetrite et al., 2021; Chen et al., 2021).

Diffusion processes are popular for generative modeling (Ho et al., 2020; Song et al., 2020), where a sample from a data distribution is gradually drifted and becomes noisier as it approaches a completely noisy sample, typically Gaussian. The celebrated paper of Song et al. (2021) suggested finding the *drift* term of the model by learning the *score* function and plugging it back into the reverse stochastic differential equation (SDE) (Anderson, 1982). As we state later, a solution to SB can be written as a *finite-energy* diffusion process. It then becomes natural to investigate diffusion processes in the context of RD theory, as generative models casting the source probability to the distortion-optimal reconstruction distribution. As we show in this work, this fresh point of view reveals surprising analytical results as well as novel estimation methods.

**In this paper**, we focus on the computation of the RD function for continuous data sources under the MSE distortion. This leads to an OT problem with quadratic cost. More specifically, we exploit the connection between RD and EOT to propose a novel stochastic control formulation to RD, where the classical result of Schrödinger (1932) implies that the tradeoff between rate and MSE distortion is equivalent to a tradeoff between the control *energy* and the *differential entropy* of the terminal state, whose probability law yields the *reconstruction* distribution. For a special class of sources, we show that the optimal control law and trajectory in the space of probability measures are given by solving a *Backward* Heat Equation (BHE). In the more general case, our approach gives rise to a numerical solution method in which the RD function is estimated using diffusion processes, with a constant diffusion coefficient.

**Our contributions:**

1. We establish a novel **connection between RD and optimal control** by presenting Terminal-Entropy Control (TEC), a stochastic-control formulation regularized by terminal uncertainty, and showing that this formulation is equivalent to the RD problem.

2. We characterize the optimal solution to TEC under certain regularity conditions.

3. Using this characterization, we provide a **closed-form solution** for the reconstruction distribution of a Gaussian-mixture source. We also demonstrate our approach on mixture distributions with non-Gaussian components via Fourier analysis. To the best of our knowledge, **such results were previously unknown**, which emphasizes the theoretical contribution of our work.

4. Based on our approach, we propose R2D2, a novel neural method for estimating the RD function and the reconstruction distributions using a simple diffusion model.

## 2 PRELIMINARIES

### 2.1 THEORETICAL BACKGROUND

**Rate-Distortion theory** Let $X \sim p_X = \mathbb{P}_0 \in \mathcal{P}(\mathbb{R}^d)$ for $d \geq 1$ denote a single sample from the source, where $\mathcal{P}(\mathbb{R}^d)$ is the set of probability measures on $\mathbb{R}^d$, and where the different source samples are *i.i.d.*. Let $\hat{X} \in \mathbb{R}^d$ denote a reconstruction sample, where the pair $(X, \hat{X})$ follows the joint distribution $(X, \hat{X}) \sim p_{X\hat{X}} \in \mathcal{P}(\mathbb{R}^d \times \mathbb{R}^d)$. In addition, let $p_{\hat{X}|X}$ denote the induced conditional distribution, which is also called the *reconstruction law*. Let $d(\cdot, \cdot) \colon \mathbb{R}^d \times \mathbb{R}^d \to \mathbb{R}_+$ denote a distortion measure between $x \in \mathbb{R}^d$ and $\hat{x} \in \mathbb{R}^d$, and let the average distortion be $D(\hat{X}, X) \triangleq \mathbb{E}[d(\hat{X}, X)]$, which is an implicit function of $p_{X\hat{X}}$. For a given pair of probability measures $\mathbb{P}_0, \mathbb{P}_1 \in \mathcal{P}(\mathbb{R}^d)$, let $D_{\mathrm{KL}}(\mathbb{P}_0||\mathbb{P}_1) \triangleq \mathbb{E}_{X \sim \mathbb{P}_0}[\log \frac{\mathrm{d}\mathbb{P}_0}{\mathrm{d}\mathbb{P}_1}(X)]$ denote the Kullback–Leibler divergence, and let the mutual information be $\mathcal{I}(X; \hat{X}) = D_{\mathrm{KL}}(p_{X\hat{X}}||p_X \otimes p_{\hat{X}})$. The lossy compression theorem (Shannon, 1959; Berger, 2003; Cover and Thomas, 2012) states that the operational RD function is equivalent to the informational RD function

$$R(D) \triangleq \min_{p_{\hat{X}|X} \colon D(\hat{X}, X) \leq D} \mathcal{I}(X; \hat{X}), \tag{1}$$

on which we will focus. The BA algorithm (Cover and Thomas, 2012) computes $R(D)$ by optimizing the Lagrangian (with a Lagrange multiplier $\epsilon > 0$)

$$L_{BA}(p_{\hat{X}|X}, \mathbb{P}_0, \epsilon) = D(\hat{X}, X) + \epsilon \mathcal{I}(\hat{X}; X) \tag{2}$$

*w.r.t.* $p_{\hat{X}|X}$. Alternatively, if we let $\mathbb{P}_1 = p_{\hat{X}}$ be the marginal distribution of the reconstruction, then an equivalent formulation is (Yang et al., 2024)

$$\min_{p_{\hat{X}|X}} L_{BA}(p_{\hat{X}|X}, \mathbb{P}_0, \epsilon) = \min_{\mathbb{P}_1 \in \mathcal{P}(\mathbb{R}^d)} \min_{(X, \hat{X}) \sim \pi \in \Pi(\mathbb{P}_0, \mathbb{P}_1)} \left\{ D(\hat{X}, X) + \epsilon D_{\mathrm{KL}}(\pi||\mathbb{P}_0 \otimes \mathbb{P}_1) \right\}, \tag{3}$$

where $\Pi(\mathbb{P}_0, \mathbb{P}_1)$ is the set of all couplings, that is, the set of joint distributions $p_{X\hat{X}} \in \mathcal{P}(\mathbb{R}^d \times \mathbb{R}^d)$ whose $X$-marginal (resp. $\hat{X}$-marginal) is $\mathbb{P}_0$ (resp. $\mathbb{P}_1$). In this work, we focus on the quadratic cost $d(\hat{x}, x) = \frac{1}{2}\|\hat{x} - x\|^2$. The average distortion is then given by $D(\hat{X}, X) \triangleq \frac{1}{2}\mathbb{E}[\|X - \hat{X}\|^2]$, and the minimization problem Eq. (3) reads

$$\min_{p_{\hat{X}|X}} L_{BA}(p_{\hat{X}|X}, \mathbb{P}_0, \epsilon) = \min_{\mathbb{P}_1} \min_{(X, \hat{X}) \sim \pi \in \Pi(\mathbb{P}_0, \mathbb{P}_1)} \left\{ \frac{1}{2}\mathbb{E}_\pi \left[\|X - \hat{X}\|^2\right] + \epsilon D_{\mathrm{KL}}(\pi||\mathbb{P}_0 \otimes \mathbb{P}_1) \right\}. \tag{4}$$

**Entropic optimal transport** For a probability measure $\mathbb{P} \in \mathcal{P}(\mathbb{R}^d)$ with density $p(x)$, $H(\mathbb{P}) \triangleq -\mathbb{E}_{X \sim \mathbb{P}} \log(p(X))$ denotes its *differential* entropy. Now, considering probability measures $\mathbb{P}_0, \mathbb{P}_1 \in \mathcal{P}(\mathbb{R}^d)$, the EOT problem (Cuturi, 2013) is given by

$$\inf_{\pi \in \Pi(\mathbb{P}_0, \mathbb{P}_1)} \left\{ \int_{\mathbb{R}^d \times \mathbb{R}^d} \frac{\|x - \hat{x}\|^2}{2} \mathrm{d}\pi(x, \hat{x}) + \epsilon D_{\mathrm{KL}}(\pi||\mathbb{P}_0 \otimes \mathbb{P}_1) \right\}. \tag{5, EOT}$$

As one may readily recognize, for values of $\epsilon$ where the optimal reconstruction has a density, the inner minimization of Eq. (4) coincides with Eq. (5, EOT) (Lei et al., 2023; Yang et al., 2024).

**The Schrödinger Bridge** The SB problem (Schrödinger, 1932) with parameter $\epsilon$ is formulated as

$$\inf_u \frac{1}{2}\mathbb{E}\left[\int_0^1 \|u(X_t, t)\|^2 \mathrm{d}t\right] \text{ s.t.} \begin{cases} X_0 \sim \mathbb{P}_0, \ X_1 \sim \mathbb{P}_1 \\ \mathrm{d}X_t = u(X_t, t)\mathrm{d}t + \sqrt{\epsilon}\mathrm{d}W_t \end{cases}, \tag{6, SB}$$

where $\mathbb{P}_0, \mathbb{P}_1$ are absolutely continuous probability measures *w.r.t.* the Lebesgue measure on $\mathbb{R}^d$, and $W_t$ is a standard Wiener process, independent of $X_0$. The *drift* $u : \mathbb{R}^d \times [0, 1] \to \mathbb{R}^d$ can be seen as a *controller* designed to steer $X_0 \sim \mathbb{P}_0$ into $X_1 \sim \mathbb{P}_1$ with *minimal average energy*, acting against a 'natural' force $W_t$. Recently, Gushchin et al. (2022) have utilized the well-established equivalence (up to an additive constant depending on $\mathbb{P}_0, \mathbb{P}_1, \epsilon$) between SB and EOT (Léonard, 2013; Chen et al., 2021), showing that the latter can be optimized via a game-theoretic formulation. The optimal joint probability $\pi$ in Eq. (5, EOT) is then given by the joint probability law of $(X_0, X_1)$.

## 2.2 RELATED WORK

**Neural estimation of information-theoretic quantities**  Following ongoing research on neural models, a variety of methods have emerged in recent years for estimating information measures, as well as for the design of compression methods aimed at achieving these fundamental limits (see the survey by Yang et al. (2023)). Belghazi et al. (2018), for example, used the Donsker–Varadhan identity to estimate MI. Kholkin et al. (2025) used samples from Brownian bridges to estimate both the MI between datasets and differential entropies. Lei et al. (2022) suggested approximating the RD function using a deep neural network model and proposed an operational coding scheme. Tsur et al. (2024) suggested approximating RD by modeling unknown input distributions, both continuous and discrete. Recently, Lei et al. (2023); Yang et al. (2024) pointed out an intriguing connection between the BA algorithm and EOT. This connection was further exploited in Yang et al. (2024) to estimate the RD function using a discrete approximation of the reconstruction probability law. Finally, Zou et al. (2025) used the connection between EOT and the Schrödinger problem to characterize this density. In this paper, we target the reconstruction distribution in RD problems with MSE loss, *i.e.*, a quadratic cost, by optimizing a *continuous* diffusion model. Compared to the methods mentioned above, this novel approach allows us not only to offer new analytic solutions to this classical problem but also to better approximate RD functions, especially for high rates, as we show in § 5.

**Diffusion models for lossy compression**  Lossy compressors that hinge on diffusion processes have become popular in recent years. In Elata et al. (2025) pre-trained diffusion models were sampled for zero-shot image compression. Recently, Ohayon et al. (2025) replaced the noise at every timestep of the reverse diffusion with samples from a sequence of predefined codebooks, achieving high perceptual reconstruction quality (Blau and Michaeli, 2019; Freirich et al., 2021; 2024). In Theis et al. (2022) a noisy version of the source data was compressed and then used in a reverse diffusion model for reconstruction. Here, we use a forward model to obtain the reconstruction distribution and compute both its rate and distortion.

**Schrödinger Bridge and EOT**  Numerical solution methods to SB include iterative fitting (Shi et al., 2023), adjoint-state matching (Liu et al., 2025; Domingo-Enrich et al., 2024) and sampling the potentials of the system (Puchkin et al., 2025a;b; Rapakoulias et al., 2024; Gushchin et al., 2024). Dai Pra (1991) further expressed the optimal control and the objective Eq. (6, SB) in terms of these potentials. This differs from our approach, where we directly optimize the control function $u$, avoiding the need to evaluate or sample potentials. The closest method to ours is given in Gushchin et al. (2022), which used the equivalence between SB and EOT (Chen et al., 2021; Léonard, 2013) to offer a game formulation for the former, which can be solved by optimizing a diffusion model. In § 3, we propose a modified problem in which the target probability is free. As such, the terminal constraint is replaced by a penalty on the final state for being uncertain. We show (Thm. 3.1) that this formulation is equivalent to the RD problem under the MSE distortion.

**Entropy-regularized stochastic control**  In addition to presenting a novel approach to the RD problem, our results can also be considered a contribution to control theory. As we stated, Eq. (6, SB) is formulated as a stochastic-control ('dynamic') problem. Both this form and its 'static' counterpart are given in Chen et al. (2021). Entropy regularization is common in stochastic control (Lambert et al., 2025) and reinforcement learning (Haarnoja et al., 2018; Ziebart et al., 2008). However, most studies aim to *maximize* the control-policy entropy, encouraging diverse actions. Alternatively, Fridman and Shaked (2000) proposed minimizing the steady-state entropy of closed-loop linear systems in $H_\infty$ control problems with infinite horizons. Here, we focus on *penalizing* the terminal-state uncertainty, leading to a novel tradeoff between energy and entropy. After submitting this manuscript for publication, we became aware of the work of Pavon (1989), which, within the context of physical systems, derived a variational form similar to Eq. (15, var-TEC). However, there, the controlled state and observation have pointwise initial conditions and control is *reversed* in time. To the best of our knowledge, our work is the first to present such a formalism in the context of information theory, which may open the door to additional applications in the broader fields of statistics and stochastic control, connecting these major disciplines.

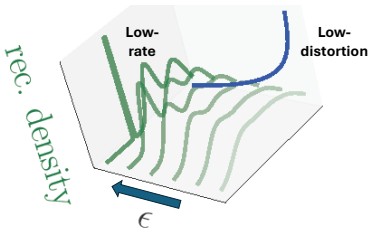

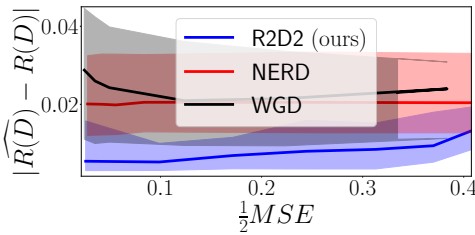

Figure 1: **Evolution of reconstruction distributions w.r.t. parameter $\epsilon$.** Typically, we assume the reconstruction density to be close to the continuous source in the low distortion regime (small $\epsilon$'s), while eventually becoming singular for a low enough rate (large $\epsilon$'s).

Figure 2: **Estimation error on a 1-D Gaussian source.** $X_0 \sim \mathcal{N}(0,1)$ and we applied R2D2 (Alg. 1) to $\epsilon \in [0.05, 0.95]$. We compare our algorithm to NERD and WGD, where we observe the improved accuracy of R2D2 over existing methods.

## 3 RATE-DISTORTION FUNCTIONS AND TERMINAL-ENTROPY STOCHASTIC CONTROL

### 3.1 PROBLEM STATEMENT

Let the source $X_0 \sim \mathbb{P}_0$, where $\mathbb{P}_0 \in \mathcal{P}_2(\mathbb{R}^d)$ is absolutely continuous *w.r.t.* the Lebesgue measure, and where $\mathcal{P}_2(\mathbb{R}^d)$ is the subset of $\mathcal{P}(\mathbb{R}^d)$ for which $\mathbb{E}\left[\|X\|^2\right] < \infty$. We further assume that $H(\mathbb{P}_0)$ is finite. As in the BA algorithm, we aim to compute the RD Lagrangian

$$\mathcal{L}_{RD}(\mathbb{P}_0, \epsilon) \triangleq \min_{\mathbb{P}_1} \min_{\pi \in \Pi(\mathbb{P}_0, \mathbb{P}_1)} \left\{ \frac{1}{2\epsilon} \int_{\mathbb{R}^d \times \mathbb{R}^d} \|\hat{x} - x\|^2 \mathrm{d}\pi(x, \hat{x}) - H(\pi) + H(\mathbb{P}_1) \right\} + H(\mathbb{P}_0), \quad \text{(7, RD)}$$

which follows from standard decomposition of MI to entropy terms (Cover and Thomas, 2012), $D_{\mathrm{KL}}(\pi||\mathbb{P}_0 \otimes \mathbb{P}_1) = H(\mathbb{P}_1) + H(\mathbb{P}_0) - H(\pi)$. Here, $\epsilon > 0$ is a tuning parameter, and $H(\mathbb{P}_0)$ is not subject to optimization.

**Assumption A1:** Eq. (7, RD) admits a solution in which the optimal reconstruction distribution is absolutely continuous and satisfies $\mathbb{P}_1 \in \mathcal{P}_2(\mathbb{R}^d)$, with finite differential entropy $H(\mathbb{P}_1)$.

Although **A1** is necessarily violated for low rates (high values of $\epsilon$), in many cases it is still expected to be valid in the low-distortion regime, where $\mathbb{P}_1$ is close to $\mathbb{P}_0$, as illustrated in Fig. 1. This holds in a variety of settings, as we demonstrate in § 3.3 § 5. Targeting the low-distortion regime is of special interest since, in modern communication, high bandwidth channels are often available (Chafii et al., 2023), and thus sources are compressed at high bitrates, allowing low distortion.

Let the process $\mathrm{d}W_t^\epsilon = \sqrt{\epsilon}\mathrm{d}W_t$, with initial state $W_0^\epsilon \sim \mathbb{P}_0$, be the (scaled) Brownian motion starting at $\mathbb{P}_0$, and denote $\pi^\epsilon = \mathbb{P}_{W_0^\epsilon, W_1^\epsilon}$, the joint law of its start and end points. It is known (Gushchin et al., 2022; Chen et al., 2021) that

$$D_{\mathrm{KL}}(\pi||\pi^\epsilon) = \frac{1}{2\epsilon} \int_{\mathbb{R}^d \times \mathbb{R}^d} \|\hat{x} - x\|^2 \mathrm{d}\pi(x, \hat{x}) - H(\pi) + H(\mathbb{P}_0) + \frac{d}{2}\log(2\pi\epsilon) \quad (8)$$

for every $\pi$ with marginal distribution $\mathbb{P}_0$. Let $\mathcal{F}(\mathbb{P}_0, \mathbb{P}_1)$ be the class of random trajectories $T_t \in \mathbb{R}^d, t \in [0, 1]$ with a joint distribution $\pi_T = \mathbb{P}_{T_0, T_1} \in \Pi(\mathbb{P}_0, \mathbb{P}_1)$. For every $T \in \mathcal{F}(\mathbb{P}_0, \mathbb{P}_1)$,

$$D_{\mathrm{KL}}(T||W^\epsilon) = D_{\mathrm{KL}}(\pi_T||\pi^\epsilon) + \int_{\mathbb{R}^d \times \mathbb{R}^d} D_{\mathrm{KL}}(T|_{x,\hat{x}}||W^\epsilon|_{x,\hat{x}})\mathrm{d}\pi_T(x, \hat{x}), \quad (9)$$

where there exists a process $T_{\mathbb{P}_1}^*$, minimizing $D_{\mathrm{KL}}(T||W^\epsilon)$ over $\mathcal{F}(\mathbb{P}_0, \mathbb{P}_1)$ with $D_{\mathrm{KL}}(T_{\mathbb{P}_1}^*|_{x,\hat{x}}||W^\epsilon|_{x,\hat{x}}) = 0$ for all $x, \hat{x} \in \mathbb{R}^d$ (Léonard, 2013, Prop. 4.1,2.3). The process $T_{\mathbb{P}_1}^*$ is known to take the form $T_{u_{\mathbb{P}_1}}^*$ of an Itô diffusion (Gushchin et al., 2022)

$$T_u : \mathrm{d}X_t = u(X_t, t)\mathrm{d}t + \sqrt{\epsilon}\mathrm{d}W_t, \quad (10)$$

with drift $u_{\mathbb{P}_1}$, where $\mathbb{E}\left[\int_0^1 \|u_{\mathbb{P}_1}(X_t, t)\|^2 \mathrm{d}t\right] < \infty$. Furthermore, for such a finite-energy process we have (Pavon and Wakolbinger, 1991),

$$D_{\mathrm{KL}}(T_{\mathbb{P}_1}^* \| W^\epsilon) = \frac{1}{2\epsilon}\mathbb{E}\left[\int_0^1 \|u_{\mathbb{P}_1}(X_t, t)\|^2 \mathrm{d}t\right]. \tag{11}$$

Considering Eq. (7, RD)-Eq. (11), we suggest the following *surrogate loss*

$$\tilde{\mathcal{L}}_{RD}(\mathbb{P}_0, \epsilon) \triangleq \min_{\mathbb{P}_1} \min_{T_u \in \mathcal{F}(\mathbb{P}_0, \mathbb{P}_1)} \left\{ \frac{1}{2\epsilon}\mathbb{E}\left[\int_0^1 \|u(X_t, t)\|^2 \mathrm{d}t\right] + H(\mathbb{P}_1) \right\} - \frac{d}{2}\log(2\pi\epsilon), \tag{12}$$

where $T_u$ is a finite-energy diffusion given by Eq. (10) with drift $u \in \mathcal{U} \triangleq \{u(x, t) : H(\mathbb{P}_1) > -\infty\}$, that is $\mathbb{P}_1$ has a finite differential entropy. The above discussion leads to the following equivalence between Eq. (7, RD) and Eq. (12).

**Theorem 3.1.** *Given $\mathbb{P}_0$ and $\epsilon > 0$, under **A1** we have $\mathcal{L}_{RD} = \tilde{\mathcal{L}}_{RD}$. Furthermore, let $u^*(x, t)$ minimize the surrogate objective*

$$u^* \in \arg\min_{u \in \mathcal{U}} \left\{ \frac{1}{2\epsilon}\mathbb{E}\left[\int_0^1 \|u(X_t, t)\|^2 \mathrm{d}t\right] + H(X_1) \right\}, \tag{13}$$

*under the law in Eq. (10). Then, the distribution $\mathbb{P}_{X_1^*}$ of $X_1^*$ associated with $u^*$ through Eq. (10) is the minimizer in Eq. (12), where $\mathbb{P}_1^* = \mathbb{P}_{X_1^*}$ is the optimal reconstruction distribution in Eq. (7, RD) and $\pi^* = \mathbb{P}_{X_0^*, X_1^*}$ is the optimal plan.*

The proof is given in App. C, where we also establish the opposite direction; whenever $(\mathbb{P}_1, \pi)$ minimizes Eq. (7, RD), there exists a drift term $u(x, t)$ minimizing Eq. (13) under Eq. (10), where $(X_0, X_1) \sim \pi$.

## 3.2 CONNECTION TO STOCHASTIC CONTROL: THE ENERGY-ENTROPY TRADEOFF

Motivated by Thm. 3.1, we present the problem of Terminal-Entropy regularized stochastic Control

$$\inf_{u \in \mathcal{U}} \left\{ \frac{1}{2}\mathbb{E}\left[\int_0^1 \|u(X_t, t)\|^2 \mathrm{d}t\right] + \epsilon H(X_1) \right\} \text{ s.t. } \begin{cases} X_0 \sim \mathbb{P}_0, \ \mathbb{P}_{X_1} \text{ is } \textit{free} \\ \mathrm{d}X_t = u(X_t, t)\mathrm{d}t + \sqrt{\epsilon}\mathrm{d}W_t \end{cases}, \tag{14, TEC}$$

where the admissible control set is again $\mathcal{U} = \{u(x, t) : H(\mathbb{P}_1) > -\infty\}$. In light of Eq. (6, SB), here $u$ can be viewed as a control law for reducing the terminal-state uncertainty while spending minimal energy. As a consequence of Thm. 3.1, in order to estimate the RD function for the source $X_0 \sim \mathbb{P}_0$, one should solve Eq. (14, TEC) with a range of $\epsilon$ values. Given the drift term $u$, $X_1$ can be efficiently sampled (*e.g.*, using the Euler–Maruyama algorithm, see App. A). Taking Assumption **A1** into account, Eq. (14, TEC) takes the following variational form

$$\inf_{u \in \mathcal{U}} \left\{ \frac{1}{2}\int_{\mathbb{R}^d} \mathrm{d}x \int_0^1 \mathrm{d}t \|u(x, t)\|^2 p_t(x) - \epsilon \int_{\mathbb{R}^d} \mathrm{d}x p_1(x)\log p_1(x) \right\}, \tag{15, var-TEC}$$

where, for the diffusion process in Eq. (10), the density state $p_t(x)$ is governed by the *Fokker-Planck* equation (Oksendal, 2013)

$$\frac{\partial}{\partial t}p_t(x) = -\nabla \cdot (p_t(x)u(x, t)) + \frac{1}{2}\epsilon\Delta_{xx}p_t(x), \ p_0(x) = \mathbb{P}_0(x), \tag{16, FPE}$$

with $\nabla\cdot = \sum \frac{\partial}{\partial x_i}e^{(i)}$ being the *divergence* operation, and $\Delta_{xx} = \sum \frac{\partial^2}{\partial x_i^2}$ being the *Laplace* operator. Interestingly, under suitable regularity conditions, the solution to Eq. (14, TEC) can be characterized by a simple equation, as we show next (proof is given in App. C):

**Theorem 3.2.** *Let $p_t^*(x) \in \mathcal{C}^{1,2}([0, 1] \times \mathbb{R}^d)$[1] satisfy the backward heat equation (BHE)*

$$\boxed{\frac{\partial}{\partial t}p_t^*(x) = -\frac{1}{2}\epsilon\Delta_{xx}p_t^*(x), \ p_0^*(x) \sim \mathbb{P}_0,} \tag{17, BHE}$$

*such that $\log p_t^*(x) \in \mathcal{C}^{1,2}([0, 1] \times \mathbb{R}^d)$ and $\|\nabla p_t^*(x)\| \log p_t^*(x) \to 0$ as $\|x\| \to \infty$ for all $t \in [0, 1]$. Let $u^* = \epsilon\nabla \log p_t^*(x)$, where $\nabla \log p_t(x)$ is the Stein score function. Then, $(u^*, p_t^*)$ is an optimal pair in Eq. (15, var-TEC)-Eq. (16, FPE) and the optimal solution to Eq. (14, TEC) admits the SDE*

$$\mathrm{d}X_t^* = \epsilon\nabla \log p_t^*(X_t)\mathrm{d}t + \sqrt{\epsilon}\mathrm{d}W_t. \tag{18}$$

---

[1]$\mathcal{C}^{1,2}([0, 1] \times \mathbb{R}^d)$ is the set of functions that are continuously differentiable *w.r.t.* $t$, and twice continuously differentiable *w.r.t.* $x$.

**Fourier analysis of Eq. (17, BHE)**   Let us assume $d = 1$ for simplicity; similar arguments hold in higher dimensions. Let the source $X_0 \sim \mathbb{P}_0$ in $\mathbb{R}$ have density $p_0$ and characteristic function $\hat{p}(\omega)$, namely $p_0(x) = \frac{1}{2\pi} \int_{-\infty}^{\infty} e^{i\omega x} \hat{p}(\omega) \mathrm{d}\omega$. It is easy to verify that a solution to Eq. (17, BHE) is

$$p_t(x) = \frac{1}{2\pi} \int_{-\infty}^{\infty} e^{i\omega x + \frac{1}{2}\epsilon\omega^2 t} \hat{p}(\omega) \mathrm{d}\omega, \tag{19}$$

whenever the integral converges for all $t \in [0, 1]$ and $\nabla \log p_t(x)$ is defined for all $x, t \in \mathbb{R} \times [0, 1]$.

### 3.3   SPECIAL CASES

Backward heat conductance problems are generally ill-posed and unstable (Miranker, 1961; Fu et al., 2007). However, Thm. 3.2 yields an exact solution for certain special cases, as we show next.

**Gaussian source**   We begin with the canonical example of a scalar Gaussian source $\mathbb{P}_0 = \mathcal{N}\left(0, \sigma_0^2\right)$, and show how our formulation recovers its known RD function. For $\epsilon < \sigma_0^2$, a solution to Eq. (17, BHE) is given by $p_t(x) = \frac{1}{\sqrt{2\pi(\sigma_0^2 - \epsilon t)}} e^{-\frac{x^2}{2(\sigma_0^2 - \epsilon t)}}$. The optimal controller is therefore given by $u(x, t) = \epsilon \nabla \log p_t(x) = -\frac{\epsilon}{\sigma_0^2 - \epsilon t} x$. Evidently, under $u(x, t)$, $X_0$ and $X_1$ are jointly Gaussian where $D = \frac{1}{2}\mathbb{E}[(X_0 - X_1)^2] = \frac{1}{2}\epsilon$, and $R = \mathcal{I}(X_0; X_1) = -\frac{1}{2}\log\left(\frac{\epsilon}{\sigma_0^2}\right)$, and we recover the known closed-form result

$$R_{\text{Gauss}}(D) = \frac{1}{2}\log\left(\frac{\sigma_0^2}{2D}\right), \quad 0 < 2D < \sigma_0^2. \tag{20}$$

Note that the factor of 2 arises because our distortion definition uses half of the MSE. This result can be easily generalized to Gaussian vector sources and $\epsilon$ values smaller than the eigenvalues of the covariance matrix $\Sigma$. In this case, the solution to Eq. (17, BHE) is $p_t(x) = \mathcal{N}(0, \Sigma - \epsilon t I)$.

**Gaussian-mixture source**   The Gaussian example can be easily extended to the case of a *Gaussian mixture*, for which no closed-form solution for the RD function is known. In this case,

$$p_0(x) = \sum_{i=1}^{N} \frac{p_i}{\sqrt{2\pi\sigma_i^2}} e^{-\frac{(x - \mu_i)^2}{2\sigma_i^2}}. \tag{21}$$

Because Eq. (17, BHE) is a linear equation, the solution is given by the superposition

$$p_t(x) = \sum_{i=1}^{N} \frac{p_i}{\sqrt{2\pi(\sigma_i^2 - \epsilon t)}} e^{-\frac{(x - \mu_i)^2}{2(\sigma_i^2 - \epsilon t)}}, \quad \epsilon \in \left(0, \min_i \sigma_i^2\right). \tag{22}$$

The optimal controller $u(x, t) = \epsilon \nabla \log p_t(x)$ is derived accordingly. Knowing $p_t(x)$ and $u(x, t)$, it is possible to obtain rate and distortion values through a Monte Carlo simulation of Eq. (14, TEC), or via neural estimation as we suggest in § 4.

**A non-Gaussian-mixture source**   To show the wide applicability of Thm. 3.2, we now apply its result to a mixture of $\text{sinc}^4$ functions. Being band-limited, for settings where $p_0(x) > 0$, such a source satisfies the conditions of the theorem and allows the desirable frequency-domain analysis of § 3.2. Although this is a toy problem, we emphasize that to the best of our knowledge, no other approach is known to tackle this case.

We consider the source $X_0$ drawn from the mixture $p_0(x) = \sum_{i=1}^{N} p_i C_i^{-1} \text{sinc}^4(\frac{x}{m_i})$ where $\text{sinc}(x) \triangleq \frac{\sin(x)}{x} \in \mathcal{C}^\infty(\mathbb{R})$, and $C_i = \frac{2\pi}{3} m_i$ are appropriate normalization factors. In this case, the characteristic function is $\hat{p}_0(\omega) = \sum_{i=1}^{N} p_i C_i^{-1} m_i \tilde{p}_0(m_i \omega)$ where $\tilde{p}_0(\omega) = \frac{1}{2\pi}\left[\pi(1 - \frac{1}{2}|\omega|)_+ * \pi(1 - \frac{1}{2}|\omega|)_+\right]$, and $*$ is the *convolution* operation. Now, for non-vanishing mixture distributions and sufficiently small values of $\epsilon$, the Fourier analysis of Eq. (19) implies that we can write the solution to Eq. (17, BHE) as

$$p_t(x) = \frac{1}{2\pi} \int_{-4}^{4} e^{i\omega x + \frac{1}{2}\epsilon\omega^2 t} \hat{p}_0(\omega) \mathrm{d}\omega = \frac{2}{2\pi} \sum_{i=1}^{N} p_i C_i^{-1} m_i \int_0^4 \tilde{p}_0(m_i \omega) \cos(\omega x) e^{\frac{1}{2}\epsilon\omega^2 t} \mathrm{d}\omega. \tag{23}$$

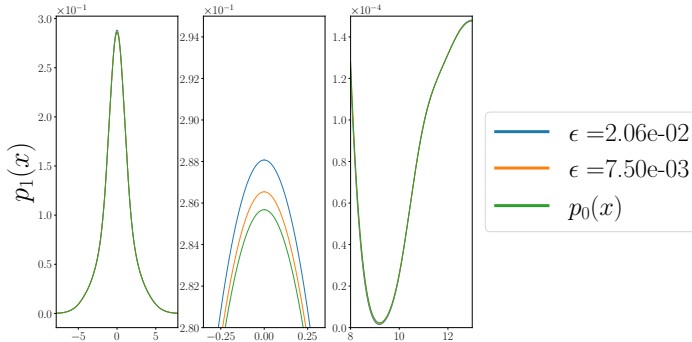

Figure 3: **Reconstruction distribution** $p_1(x)$ **of non-Gaussian mix source.** In the **(left)** pane, we approximate reconstruction distribution Eq. (23) by numerical integration. A closer look at different points is provided in the **(middle)** and **(right)** panes.

Fig. 3 demonstrates this result for $N = 4, m_i = \left[1, \sqrt{2}, \pi, e\right]$ and $p_i = \frac{1}{4}$, where we numerically integrated Eq. (23) to approximate the reconstruction distribution $p_1(x)$ for different values of $\epsilon$. Parameters were chosen such that $p_t(x) > 0$ everywhere on $\mathbb{R}$.

## 4 R2D2: NEURAL ESTIMATION OF RATE-DISTORTION FUNCTIONS

Although elegant, as may be pointed out, the assumptions of Thm. 3.2 may be restrictive: They require $p_t(x)$ to be non-vanishing and twice differentiable everywhere in $\mathbb{R}^d$; even under these requirements, Eq. (17, BHE) may be ill-posed, depending on the initial condition $\mathbb{P}_0$; Finally, the explicit source distribution is rarely known in practice, and is instead accessible only from samples. Therefore, in this section, we propose R2D2 (Alg. 1), a sample-based method for solving Eq. (14, TEC) and approximating the RD function in the general case (under Assumption **A1**).

---

**Algorithm 1** Revealing RD functions with Diffusion (R2D2)

---

1: **Input:** source $X_0 \sim \mathbb{P}_0 \in \mathcal{P}(\mathbb{R}^d)$, initial controller $u_\theta$, batch size $M$, timestep $\Delta_t, \epsilon_{\min}, \epsilon_{\max} > 0$, learning rate $\alpha$.
2: **while** Training **do**                                                    ▷ Training
3:     Choose $\epsilon \sim \text{Uniform}[\epsilon_{\min}, \epsilon_{\max}]$.
4:     Sample batch $\{X_0^m\}_{m=1}^M \sim \mathbb{P}_0$.
5:     Sample trajectory $\{u_\theta(X_{t_i}^m, t_i, \epsilon), X_1^m\}_{m=1}^M \leftarrow \text{EuMa}(u_\theta, \{X_0^m\}_{m=1}^M, \epsilon, \Delta_t)$.
6:     Estimate energy $L_\theta^\epsilon \leftarrow \frac{1}{2M} \sum_m \sum_{t_i} \|u_\theta(X_{t_i}^m, t_i, \epsilon)\|^2 \Delta_t$.
7:     Estimate terminal entropy $\hat{H}(X_1)$ (see App. B).
8:     RD loss $\mathcal{L}_\theta^\epsilon \leftarrow L_\theta^\epsilon + \epsilon \hat{H}(X_1)$.
9:     Step $\theta \leftarrow \theta - \alpha \nabla_\theta \mathcal{L}_\theta^\epsilon$.
10: **end while**
11:
12: Sample batch $\{X_0^m\}_{m=1}^M \sim \mathbb{P}_0$.                          ▷ Evaluate specific $\epsilon \in [\epsilon_{\min}, \epsilon_{\max}]$
13: Sample trajectory $\{u_\theta(X_{t_i}^m, t_i, \epsilon), X_1^m\}_{m=1}^M \leftarrow \text{EuMa}(u_\theta, \{X_0^m\}_{m=1}^M, \epsilon, \Delta_t)$.
14: Obtain RD loss $\mathcal{L}_\theta^\epsilon$ (lines **6-8**).
15: Estimate **distortion**: $\hat{D} = \frac{1}{2M} \sum_{m=1}^M \|X_1^m - X_0^m\|^2$.
16: Estimate **rate**: $\hat{R} = \frac{\mathcal{L}_\theta^\epsilon - \hat{D}}{\epsilon} - \frac{d}{2} \log(2\pi\epsilon)$.
17: **Output:** $(\hat{R}, \hat{D})$.

---

**Our method** R2D2 (summarized in Algorithm 1) is based on modeling the controller function $u_\theta(x, t, \epsilon)$ using a DNN with parameters $\theta$. The flexibility and generalizability offered by DNNs allow us to capture multiple positions on the RD-curve (different $\epsilon$ values) using a single controller model (*cf.* Yang et al. (2024)). To train our model, we access the data source $X_0$ to draw a batch of $M$ samples. Using the Euler–Maruyama method (EuMa, Alg. 2 in App. A), we sample discretized

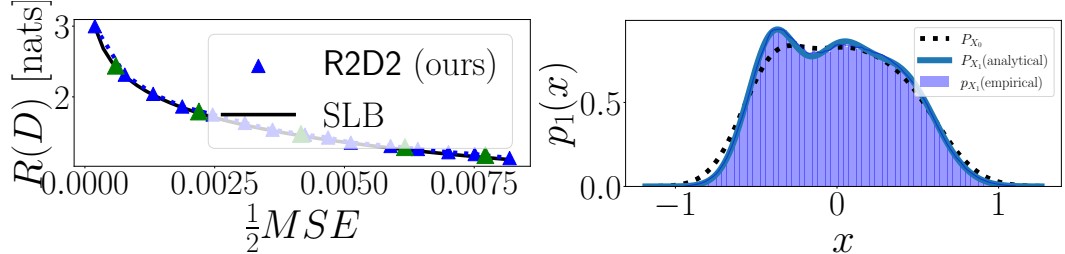

Figure 4: **The $R(D)$ function for a mixture of Gaussians.** $X_0 \sim \mathbb{P}_0$ is a mixture of Gaussians. (**left**) We apply R2D2 (Alg. 1) to $\epsilon \in [4 \times 10^{-4}, 1.64 \times 10^{-2}]$ and compare the result with SLB. *Green* markers indicate higher precision. (**right**) For $\epsilon = 1.56 \times 10^{-2}$, we plot the reconstruction distribution $\mathbb{P}_1$. The empirical distribution matches the analytical result (*bold* line).

random trajectories $X_{t_i}$ from Eq. (10). The minimization objective in Eq. (12) is approximated (up to an additive factor of $\frac{d}{2} \log(2\pi\epsilon)$) by $\mathcal{L}_\theta^\epsilon = L_\theta^\epsilon + \epsilon \hat{H}(X_1)$, where the estimated controller energy is $L_\theta^\epsilon \approx \frac{1}{2M} \sum_{m=1}^M \sum_{t_i} \|u_\theta(X_{t_i}^m, t_i, \epsilon)\|^2 \Delta_t$. The terminal entropy $\hat{H}(X_1)$ is estimated through the approximated negative entropy or through a kernel method (see App. B for details). To evaluate $R(D)$, after training, we recalculate $\mathcal{L}_\theta^\epsilon$, and use Eq. (12) to compute the empirical values

$$\hat{D}(\epsilon) = \frac{1}{2M} \sum_{m=1}^M \|X_1^m - X_0^m\|^2, \quad \hat{R}(\epsilon) = \hat{\mathcal{I}}(X_0^m; X_1^m) = \frac{\mathcal{L}_\theta^\epsilon - \hat{D}(\epsilon)}{\epsilon} - \frac{d}{2} \log(2\pi\epsilon). \quad (24)$$

**Remark 4.1.** *Both methods of Yang et al. (2024) and Lei et al. (2022) assume an upper bound on the rate (i.e., $R < \log M$, where $M$ is the batch size or the support size of the atomic probability model). Our method does not suffer from this substantial limitation. This makes Alg. 1 suitable for estimating the RD function at the low-distortion regime ('high-resolution', high rates).*

## 5 NUMERICAL RESULTS

In this section, we apply our results to both toy and real-world problems. For full details and more numerical results, we refer the reader to App. D. For full simulation details, we refer to App. E.

### 5.1 GAUSSIAN DATA

In Fig. 2, we demonstrate the efficiency of Alg. 1 on the 1-D Gaussian case[2] of § 3.3. We compare our method with NERD (Lei et al., 2022) and WGD (Yang et al., 2024) over $64$ independent experiments (seeds) and plot median absolute error with interquartile ranges. We observe that R2D2 clearly has lower estimation error then exisiting methods, in both the high-rate and low-rate regimes.

In Fig. 4, we show results for a Gaussian mixture Eq. (21) source, where $N = 3$, $\mu_i = \{-.4, 0, .4\}$, $\sigma_i^2 = \{4, 5, 6\} \times 10^{-2}$, $p_i = \frac{1}{3}$. We apply Alg. 1 to $\epsilon \in [4 \times 10^{-4}, 1.64 \times 10^{-2}]$ and compare the estimated RD function with the approximate Shannon's lower bound (SLB) (Cover and Thomas, 2012; Berger, 2003), given here by $H(\mathbb{P}_0) - \frac{1}{2} \log(4\pi e D)$. For $\epsilon = 1.56 \times 10^{-2}$, we further plot the reconstruction distribution $\mathbb{P}_1$, the law of the diffusion process's outcome. We observe that the empirical distribution obtained by Alg. 1 matches the closed-form in Eq. (22).

### 5.2 REAL DATA

#### 5.2.1 CIFAR10 DATASET

We demonstrate the efficiency of Alg. 1 on a realistic source. More specifically, as input to R2D2, we sample $4 \times 4$-pixel grayscale image patches from the 'CIFAR10' dataset (Krizhevsky and Hinton,

---

[2]All our codes are publicly available at https://github.com/ML-group-il/r2d2.

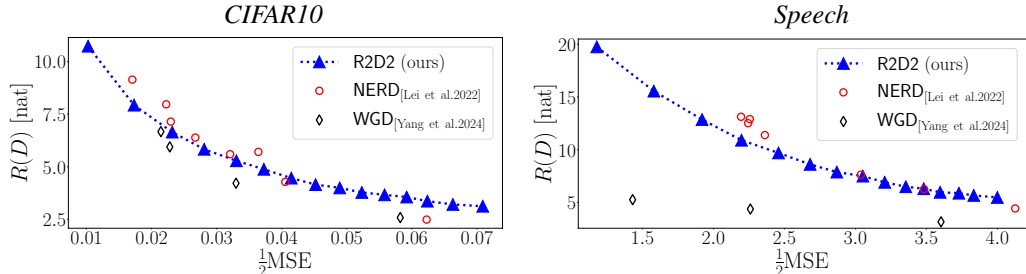

Figure 5: **Estimating $R(D)$ functions on real data. (left) CIFAR10 images.** $X_0$ is a random $4 \times 4$ patch from a grayscale image. The RD function, as estimated by R2D2 (Alg. 1) and baselines. **(right) Speech dataset.** We observe the efficiency of R2D2 at high-rates ($> 20\,[\text{nats}]$), while existing methods are practically upper bounded by $\sim 13\,[\text{nats}]$.

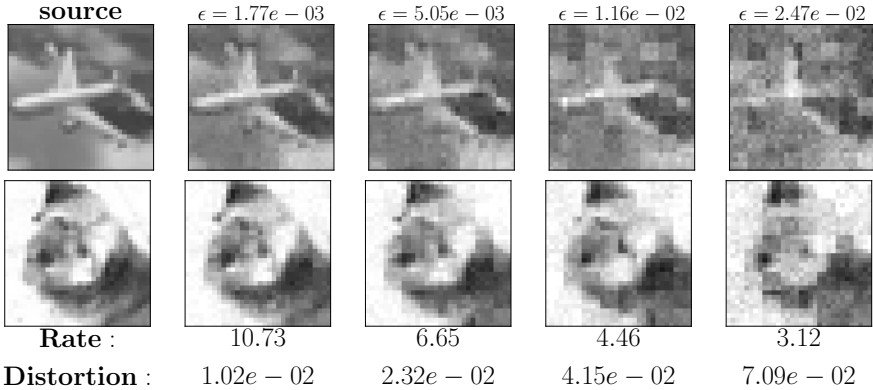

Figure 6: **Reconstruction distribution for the CIFAR10 dataset.** Patches drawn from the reconstruction distribution $X_1$ obtained for different $\epsilon$ values.

2009). Fig. 5**(left)** demonstrates the efficiency of our method in solving this problem, where we present the RD function, as estimated by R2D2 (Alg. 1). In Fig. 6, we present images, drawn from the reconstruction distribution $\mathbb{P}_1$ for different values of $\epsilon$.

### 5.2.2 SPEECH DATASET

We further test our method on the high-dimensional Free Spoken Digit dataset of Jackson et al. (2018) (see also Yang et al. (2023; 2024)). Here, the 33-dimensional samples consist of spoken-digit recordings. The data are obtained and preprocessed as in Yang et al. (2023, § A.5.4), and each feature is then whitened. We present the estimated RD function, compared to the NERD and WGD estimations, with special attention given to approximating the low-distortion regime (see Remark 4.1). We set the latent dimension size in NERD to 1024, the batch size to $M = 1 \times 10^6$, and the number of particles in WGD to $n = 2 \times 10^5$, as in Yang et al. (2024). As we can see in Fig. 5**(right)**, our method is capable of approximating (theoretically) unbounded rate values ($> 20\,[\text{nats}]$), whereas previous methods are practically bounded by approximately $13\,[\text{nats}]$ or less.

## 6 CONCLUSION

We considered the computation of the RD function and optimal reconstruction distribution for continuous data sources under the MSE distortion. We exploited the connection between RD and EOT to estimate the RD function using diffusion processes through a novel control formulation in which the RD tradeoff is equivalent to a tradeoff between energy and entropy. Under regularity conditions, the optimal control is given by a BHE. We demonstrated our results in certain special cases, obtaining closed-form solutions, and in a real-world setting using a numerical method. This work paves the way for solving RD in settings beyond the MSE loss and continuous distributions.

ACKNOWLEDGMENTS

The work of NW was supported by the Israel Science Foundation (ISF), grant no. 1782/22.

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

## A  EULER-MARUYAMA SAMPLING

Here we review the *Euler-Maruyama* (EuMa) Algorithm, which we used in our simulations to sample from the diffusion process Eq. (10). The sampling procedure is given in Alg. 2.

---

**Algorithm 2** Euler-Maruyama (EuMa)

---

1: **Input:** $\epsilon > 0$, drift $u(x, t, \epsilon)$, initial batch $\{X_0^m\}_{m=1}^M \sim \mathbb{P}_0$, timestep $\Delta_t$.
2: **for** $t_i = 0, \Delta_t, \ldots, 1 - \Delta_t$ **and** $m = 1, \ldots, M$ **do**
3:     Sample $z_i^m \sim \mathcal{N}(0, I)$.
4:     $X_{t_{i+1}}^m \leftarrow X_{t_i}^m + u(X_{t_i}^m, t_i, \epsilon)\Delta_t + \sqrt{\epsilon \Delta_t} z_i^m$.
5: **end for**
6: Return $\{u(X_{t_i}^m, t_i, \epsilon), X_1^m\}_{m=1}^M$.

---

## B  ENTROPY ESTIMATION

For the sake of completeness of Alg. 1, here we present the techniques used in the paper for estimating entropy (line **9** in the Algorithm). We emphasize, though, that we use the entropy estimator as a black-box, where any method could be plugged-in, orthogonally to our main ideas.

In our experiments on low-dimensional settings we used the approximated negative entropy method, for its simplicity and ease of compute. This could be hardly scaled to higher dimensions since it requires the computation of large covariance matrices. For real-world settings we used the scalable kernel method of Pichler et al. (2022).

### B.1  NEGENTROPY

Negative entropy, or *negentropy* (Oja and Hyvarinen, 2000) of a random variable $X_1 \in \mathbb{R}^d$ is the difference in entropy from the Gaussian distribution with the same second-order statistics. Explicitly

$$\text{negentropy}(X_1) \triangleq H(Z) - H(X_1) = -H(X_1) + \frac{1}{2}\sum_{i=1}^d \log \lambda_i + \frac{d}{2}\log(2\pi e), \qquad (25)$$

where $\lambda_i$ are the eigenvalues of the covariance matrix $\Sigma_{X_1}$. We have the following connection between negentropy and KL divergence

**Lemma B.1.** *(Kholkin et al., 2025, Corollary A.3) Let $Z \sim \mathcal{N}(\mu_1, \Sigma_1)$ where $\mu_1, \Sigma_1$ are the mean and covariance of $X_1$, respectively. Then,*

$$H(X_1) = H(Z) - D_{\text{KL}}(\mathbb{P}_{X_1} || \mathbb{P}_Z) \qquad (26)$$

Using Eq. (26), we approximate the negentropy through the *Donsker–Varadhan* identity (Belghazi et al., 2018)

$$\text{negentropy}(X_1) = D_{\text{KL}}(\mathbb{P}_{X_1} || \mathbb{P}_Z) = \sup_f \left[ \mathbb{E}_{z \sim \mathbb{P}_{X_1}} f(z) - \log \mathbb{E}_{z \sim \mathbb{P}_Z} e^{f(z)} \right], \qquad (27)$$

which can be estimated from samples (*cf.* Kholkin et al. (2025, § A.2) and Franzese et al. (2023, § 3.2)).

In our simulations, we model the argument in Eq. (27) as a parametric model $Z_\omega(\cdot, \epsilon)$, where now we approximate the negentropy from $M$ samples of $X_1$ as

$$\text{negentropy}(X_1, \epsilon) \approx \frac{1}{M}\sum_{m=1}^M Z_\omega(X_1^m, \epsilon) - \log\left[ \frac{1}{M}\sum_{m=1}^M e^{Z_\omega(z^m, \epsilon)} \right]. \qquad (28)$$

$z^m$ are i.i.d. Gaussian samples with the empirical mean and covariance of $X_1$. The entropy $H(X_1)$ can now be estimated through Eq. (25).

## B.2 KNIFE

The KNIFE estimator (Pichler et al., 2022) is a Gaussian-mixture plug-in estimator for differential entropies. For $x \in \mathbb{R}^d$ the empirical distribution is approximated using

$$\hat{p}_{\text{KNIFE}}(x; \theta) = \Sigma_{k=0}^{K-1} u_k g_{\text{pdf}}(x; \mu_k, A_k), \tag{29}$$

where $g_{\text{pdf}}(\cdot; \mu_k, A_k)$ are Gaussian kernels with mean and variance $\mu_k, A_k$ respectively, and $u_k > 0$ are weights, $\Sigma_k u_k = 1$.

The plug-in estimation is then given by

$$\hat{H}(X_1; \theta) = -\mathbb{E}_{x \sim \mathbb{P}_1} \log\left[\hat{p}_{\text{KNIFE}}(x; \theta)\right] \geq H(X_1). \tag{30}$$

For tight estimation, $\hat{H}(X_1; \theta)$ is minimized over $\theta := \{(u_k, \mu_k, A_k)\}_{k=0}^{K-1}$.

# C PROOFS OF THINGS

## C.1 PROOF OF THM. 3.1

**Theorem C.1.** *(Thm. 3.1 in the Main text)* $\mathcal{L}_{RD} = \tilde{\mathcal{L}}_{RD}$. *Furthermore, let $u^*(x,t)$ minimize the surrogate objective*

$$u^* \in \arg\min_u \left\{ \mathbb{E}\left[\frac{1}{2\epsilon} \int_0^1 \|u(X_t,t)\|^2 \mathrm{d}t\right] + H(X_1) \right\}, \tag{31}$$

*under*

$$\mathrm{d}X_t = u(X_t,t)\mathrm{d}t + \sqrt{\epsilon}\mathrm{d}W_t, \quad X_0 \sim \mathbb{P}_0. \tag{32}$$

*Then, the distribution $\mathbb{P}_{X_1^*}$ of $X_1^*$ associated with $u^*$ through Eq. (32) is the minimizer in Eq. (12), where $\mathbb{P}_1^* = \mathbb{P}_{X_1^*}$ is the optimal reconstruction distribution in Eq. (7, RD) and $\pi^* = \mathbb{P}_{X_0^*, X_1^*}$ is the optimal plan.*

Here, we also prove the opposite direction; whenever $(\mathbb{P}_1, \pi)$ minimizes Eq. (7, RD), there exists a drift term $u(x,t)$ minimizing Eq. (31) under Eq. (32), where $X_0, X_1 \sim \pi$.

*Proof.* Let $(\mathbb{P}_1, \pi)$ be a solution to Eq. (7, RD), where $\pi \in \Pi(\mathbb{P}_0, \mathbb{P}_1)$. Then, we choose

$$T(\omega) = \int_{x,\hat{x}} W^\epsilon|_{x,\hat{x}}(\omega)\mathrm{d}\pi(x,\hat{x}) \tag{33}$$

as a probability law in $\mathcal{F}(\mathbb{P}_0, \mathbb{P}_1)$. It can be easily deduced that

$$D_{\mathrm{KL}}(T|_{x,\hat{x}}||W^\epsilon|_{x,\hat{x}}) = 0, \tag{34}$$

hence, from Eq. (8)-Eq. (9) we have

$$D_{\mathrm{KL}}(T||W^\epsilon) = D_{\mathrm{KL}}(\pi||\pi^{W^\epsilon}) = \mathbb{E}_\pi\left[\frac{\|X - \hat{X}\|^2}{2\epsilon}\right] - H(\pi) + \frac{d}{2}\log(2\pi\epsilon). \tag{35}$$

which is equal to Eq. (7, RD) up to an additive constant, depends only on $\mathbb{P}_0, \mathbb{P}_1$ and $\epsilon$. Since $\pi$ minimizes Eq. (7, RD) for $\mathbb{P}_1$, it minimizes $D_{\mathrm{KL}}(\pi||\pi^{W^\epsilon})$ over $\Pi(\mathbb{P}_0, \mathbb{P}_1)$, thus $T$ minimizes $D_{\mathrm{KL}}(T||W^\epsilon)$ over $\mathcal{F}(\mathbb{P}_0, \mathbb{P}_1)$ (see Chen et al. (2021, Problem 4.2, Eqn. (4.8)); *mutatis mutandis*). As a solution to $\min_{T \in \mathcal{F}(\mathbb{P}_0, \mathbb{P}_1)} D_{\mathrm{KL}}(T||W^\epsilon)$, $T$ takes the form Eq. (10) with some drift function $u$ (Léonard, 2013, Prop. 4.1).

Using Eq. (11) we get

$$\tilde{\mathcal{L}}_{RD} \leq \frac{1}{2\epsilon}\mathbb{E}\left[\int_0^1 \|u(X_t,t)\|^2\mathrm{d}t\right] + H(\mathbb{P}_1) - \frac{d}{2}\log(2\pi\epsilon) \tag{36}$$

$$= D_{\mathrm{KL}}(T||W^\epsilon) + H(\mathbb{P}_1) - \frac{d}{2}\log(2\pi\epsilon) \tag{37}$$

$$= D_{\mathrm{KL}}(\pi||\pi^\epsilon) + H(\mathbb{P}_1) - \frac{d}{2}\log(2\pi\epsilon) \tag{38}$$

$$= \frac{1}{2\epsilon}\mathbb{E}_\pi\left[\|X - \hat{X}\|^2\right] - H(\pi) + H(\mathbb{P}_0) + H(\mathbb{P}_1) \tag{39}$$

$$= \mathcal{L}_{RD}, \tag{40}$$

implying that $\mathcal{L}_{RD} \geq \tilde{\mathcal{L}}_{RD}$.

On the other hand, let $u^*$ as in Eq. (31), and let $\mathbb{P}_1^* = \mathbb{P}_{X_1^*}$. Clearly, $\mathbb{P}_{X_1^*}, u^*$ minimize Eq. (12). Furthermore, the law $T^*$ induced by $u^*$,

$$T^* : \mathrm{d}X_t^* = u^*(X_t^*,t)\mathrm{d}t + \sqrt{\epsilon}\mathrm{d}W_T, \quad X_0^* \sim \mathbb{P}_0, \tag{41}$$

minimizes $D_{\mathrm{KL}}(T^*||W^\epsilon) = \frac{1}{2\epsilon}\mathbb{E}\left[\int_0^1 \|u^*(X_t^*,t)\|^2\mathrm{d}t\right]$ over $\mathcal{F}(\mathbb{P}_0, \mathbb{P}_1^*)$, hence it is a solution to $\min_{T \in \mathcal{F}(\mathbb{P}_0, \mathbb{P}_1^*)} D_{\mathrm{KL}}(T||W^\epsilon)$. Thus, according to Léonard (2013, Prop. 2.3),

$D_{\mathrm{KL}}(T^*|_{x,\hat{x}}||W^\epsilon|_{x,\hat{x}}) = 0, \forall x, \hat{x}$. Let $\pi^* = \mathbb{P}_{X_0^*, X_1^*} \in \Pi(\mathbb{P}_0, \mathbb{P}_1^*)$. We have from Eq. (9)

$$D_{\mathrm{KL}}(\pi^*||\pi^\epsilon) = \mathbb{E}_{\pi^*}\left[\frac{\|X - \hat{X}\|^2}{2\epsilon}\right] - H(\pi^*) + H(\mathbb{P}_0) + \frac{d}{2}\log(2\pi\epsilon) \tag{42}$$

$$= D_{\mathrm{KL}}(T^*||W^\epsilon) \tag{43}$$

$$= \frac{1}{2\epsilon}\mathbb{E}\left[\int_0^1 \|u^*(X_t^*, t)\|^2 \mathrm{d}t\right]. \tag{44}$$

Thus,

$$\mathcal{L}_{RD} \leq \mathbb{E}_{\pi^*}\left[\frac{\|X - \hat{X}\|^2}{2\epsilon}\right] - H(\pi^*) + H(\mathbb{P}_1^*) + H(\mathbb{P}_0) \tag{45}$$

$$= \frac{1}{2\epsilon}\mathbb{E}\left[\int_0^1 \|u^*(X_t^*, t)\|^2 \mathrm{d}t\right] + H(\mathbb{P}_1^*) - \frac{d}{2}\log(2\pi\epsilon) \tag{46}$$

$$= \tilde{\mathcal{L}}_{RD}, \tag{47}$$

yielding $\mathcal{L}_{RD} = \tilde{\mathcal{L}}_{RD}$, which completes the proof. Note that arguments similar to Eq. (45)–Eq. (47) yield that under **A1**, $\tilde{\mathcal{L}}_{RD} > -\infty$. $\qquad\square$

## C.2 PROOF OF THM. 3.2

**Theorem C.2.** *(Thm. 3.2 in the Main text) Let $p_t^*(x)$ such that $\log p_t^*(x) \in \mathcal{C}^{1,2}([0,1] \times \mathbb{R}^d)$ and $\|\nabla p_t^*(x)\| \log p_t^*(x) \to 0$ as $\|x\| \to \infty$ for all $t \in [0,1]$, satisfying the BHE*

$$\frac{\partial}{\partial t}p_t(x) = -\frac{1}{2}\epsilon\Delta_{xx}p_t(x), \ p_0(x) \sim \mathbb{P}_0, \tag{48, BHE}$$

*and let $u^* = \epsilon\nabla\log p_t^*(x)$ where $\nabla\log p_t(x)$ is the Stein score function. Then, $(u^*, p_t^*)$ is an optimal pair in Eq. (15, var-TEC)-Eq. (16, FPE) and the solution to Eq. (14, TEC) admits the SDE*

$$\mathrm{d}X_t = \epsilon\nabla\log p_t^*(X_t)\mathrm{d}t + \sqrt{\epsilon}\mathrm{d}W_t. \tag{49}$$

*Proof.* Let us define the Lagrangian

$$L(u, p, \mu, \lambda) = \frac{1}{2}\int_{\mathbb{R}^d}\mathrm{d}x\int_0^1\mathrm{d}t\|u(x,t)\|^2 p_t(x)$$
$$+ \int_{\mathbb{R}^d}\mathrm{d}x\int_0^1\mathrm{d}t\mu(x,t)\left(\dot{p}_t(x) + \nabla\cdot(up(x,t)) - \frac{1}{2}\epsilon\Delta_{xx}p_t(x)\right)$$
$$- \epsilon\int_{\mathbb{R}^d}\mathrm{d}x p_1(x)\log p_1(x) + \int_{\mathbb{R}^d}\mathrm{d}x\lambda(x)(p_0(x) - \mathrm{d}\mathbb{P}_0(x)), \tag{50}$$

where $\mathrm{d}\mathbb{P}_0$ denote the *density* function of $\mathbb{P}_0$.

We now apply the following integration by parts property of the divergence: For $g : \mathbb{R}^d \to \mathbb{R}$ and $f : \mathbb{R}^d \to \mathbb{R}^d$, and a bounded domain $D \subseteq \mathbb{R}^d$ with boundary $\partial D$,

$$\int_D g(x)\nabla\cdot f(x)\mathrm{d}x = -\int_D \nabla g(x)\cdot f(x)\mathrm{d}x + \oint_{\partial D} g(x)f(x)\cdot\vec{n}\mathrm{d}a \tag{51}$$

where $\nabla, \nabla\cdot$ are the gradient and divergence operators, respectively. If $\|g(x)f(x)\|$ decays as $\|x\| \to \infty$, we can integrate over domains with large enough diameters, thus ignoring the boundary term and practically integrate over $\mathbb{R}^d$. In our setting, $p, \mu$ are scalar functions, and $u$ is a field.

Integrating by parts, we have

$$\int_0^1\mathrm{d}t\mu(x,t)\dot{p}_t(x) = \mu(x,1)p_1(x) - \mu(x,0)p_0(x) - \int_0^1\mathrm{d}t\dot{\mu}(x,t)p_t(x) \tag{52}$$

and

$$\int_{\mathbb{R}^d} \mathrm{d}x \mu(x,t) \nabla \cdot (up(x,t)) = - \int_{\mathbb{R}^d} \mathrm{d}x p u \cdot \nabla \mu(x,t) + \oint \mu p u \cdot \vec{n} \mathrm{d}a \tag{53}$$

as well as

$$\int_{\mathbb{R}^d} \mathrm{d}x \mu(x,t) \Delta_{xx} p(x,t)$$

$$= \int_{\mathbb{R}^d} \mathrm{d}x \mu(x,t) \nabla \cdot \nabla p(x,t) \tag{54}$$

$$= - \int_{\mathbb{R}^d} \mathrm{d}x \nabla \mu(x,t) \cdot \nabla p(x,t) + \oint \mu(\nabla p(x,t)) \cdot \vec{n} \mathrm{d}a \tag{55}$$

$$= - \int_{\mathbb{R}^d} \mathrm{d}x \nabla \mu(x,t) \cdot \nabla p(x,t) + \oint p \nabla \mu \cdot \vec{n} \mathrm{d}a + \oint [\mu(\nabla p(x,t)) - p \nabla \mu] \cdot \vec{n} \mathrm{d}a \tag{56}$$

$$= \int_{\mathbb{R}^d} \mathrm{d}x \nabla_{xx} \mu(x,t) p_t(x) - \oint p \nabla \mu \cdot \vec{n} \mathrm{d}a + \oint \mu(\nabla p(x,t)) \cdot \vec{n} \mathrm{d}a. \tag{57}$$

Provided that all boundary terms vanish as $\|x\| \to \infty$, and putting everything back together, we obtain

$$L(u,p,\mu,\lambda) = \frac{1}{2} \int_{\mathbb{R}^d} \mathrm{d}x \int_0^1 \mathrm{d}t \|u(x,t)\|^2 p_t(x)$$

$$- \int_0^1 \mathrm{d}t \int_{\mathbb{R}^d} \mathrm{d}x \left[ \dot{\mu}(x,t) + u(x,t) \cdot \nabla \mu(x,t) + \frac{1}{2} \epsilon \Delta_{xx} \mu(x,t) \right] p_t(x)$$

$$+ \int_{\mathbb{R}^d} \mathrm{d}x (\mu(x,1) - \epsilon \log p_1(x)) p_1(x)$$

$$- \int_{\mathbb{R}^d} \mathrm{d}x (\mu(x,0) - \lambda(x)) p_0(x) - \int_{\mathbb{R}^d} \mathrm{d}x \lambda(x) \mathrm{d}\mathbb{P}_0(x). \tag{58}$$

Taking the first variation to zero, $\frac{\delta L}{\delta u} = 0$ yields

$$u^*(x,t) = \nabla \mu(x,t). \tag{59}$$

From $\frac{\delta L}{\delta p} = 0$ we obtain (*Hamilton–Jacobi equation*),

$$\dot{\mu}(x,t) + \frac{1}{2} \|\nabla \mu(x,t)\|^2 + \frac{1}{2} \epsilon \Delta_{xx} \mu(x,t) = 0, \ \mu(x,1) = \epsilon(1 + \log p_1(x)). \tag{60}$$

We also know that (*Fokker–Planck equation*)

$$\dot{p}_t(x) + \nabla \cdot (p_t(x) \nabla \mu(x,t)) - \frac{1}{2} \epsilon \Delta_{xx} p_t(x) = 0, p_0(x) = \mathrm{d}\mathbb{P}_0. \tag{61}$$

We now substitute a solution of the form $\mu(x,t) = \epsilon(1 + \log p_t(x))$ into Eq. (60), and verify it satisfies our equations:

$$\dot{p}_t/p_t + \frac{1}{2} \epsilon \|\nabla p_t/p_t\|^2 + \frac{1}{2} \epsilon \nabla \cdot (\nabla p_t/p_t)$$

$$= \dot{p}_t/p_t + \frac{1}{2} \epsilon \|\nabla p_t/p_t\|^2 + \frac{1}{2} \epsilon \sum_i (p_{x_i x_i}/p_t - (p_{x_i}/p_t)^2) \tag{62}$$

$$= \dot{p}_t/p_t + \frac{1}{2} (\epsilon - \epsilon) \|\nabla p_t/p_t\|^2 + \frac{1}{2} \epsilon \Delta_{xx} p_t/p_t = 0 \tag{63}$$

where the last equality stems from Eq. (48, BHE). $\qquad \square$

# D    EXTENDED RESULTS AND TECHNICAL DETAILS

For the sake of completeness and in-depth reading, in this section we extend § 5 with full technical details and additional results. We also present detailed and full-sized figures for improved accessibility.

## D.1    GAUSSIAN SOURCES

Let $\mathbb{P}_0 = \mathcal{N}\left(0, \sigma_0^2\right)$. A solution to Eq. (17, BHE) is given by

$$p_t(x) = \frac{1}{\sqrt{2\pi(\sigma_0^2 - \epsilon t)}} e^{-\frac{x^2}{2(\sigma_0^2 - \epsilon t)}}. \tag{64}$$

The optimal controller is hence given by

$$u(x,t) = \epsilon \nabla \log p_t(x) = -\frac{\epsilon}{\sigma_0^2 - \epsilon t} x. \tag{65}$$

Let us denote $a_t = \frac{\epsilon}{\sigma_0^2 - \epsilon t}$ and $r = \frac{\epsilon}{\sigma_0^2}$. It is easy to see that under $u(x,t)$, $X_0$ and $X_1$ are jointly-Gaussian where

$$\mathrm{d}X_t = -a_t X_t \mathrm{d}t + \sqrt{\epsilon}\mathrm{d}W_t \tag{66}$$

$$U_t^{-1} X_t = X_0 + \sqrt{\epsilon} \int_0^t U_s^{-1} \mathrm{d}W_s, \tag{67}$$

with

$$U_t = e^{-\int_0^t a_s \mathrm{d}s} = 1 - rt. \tag{68}$$

That is,

$$X_1 = (1 - r)X_0 + N_1 \tag{69}$$

$$\sigma_{N_1}^2 = \epsilon(1 - r)^2 \int_0^1 U_s^{-2} \mathrm{d}s = \epsilon(1 - r)^2 \frac{\sigma_0^2}{\sigma_0^2 - \epsilon} = \epsilon(1 - r). \tag{70}$$

where $N_1$ is a Gaussian noise, independent of $X_0$. We can now compute the distortion

$$\mathbb{E}\left[(X_1 - X_0)^2\right] = r^2 \sigma_0^2 + \epsilon(1 - r) = \epsilon - \epsilon^2 \sigma_0^{-2} + \epsilon^2 \sigma_0^{-2} = \epsilon, \tag{71}$$

and the correlation

$$\rho \triangleq \mathbb{E}\left[X_0 X_1\right] = (1 - r)\sigma_0^2 = \sigma_0^2 - \epsilon \tag{72}$$

and then also compute the MI by plugging

$$\sigma_1^2 = (1 - r)^2 \sigma_0^2 + \epsilon(1 - r) = \sigma_0^2 - 2\epsilon + \epsilon^2 \sigma_0^{-2} + \epsilon - \epsilon^2 \sigma_0^{-2} = \sigma_0^2 - \epsilon \tag{73}$$

into

$$\mathcal{I}(X_0; X_1) = -\frac{1}{2}\log\left(1 - \frac{\rho^2}{\sigma_1^2 \sigma_0^2}\right) \tag{74}$$

$$= -\frac{1}{2}\log\left(1 - \frac{(1 - r)^2 \sigma_0^4}{(\sigma_0^2 - \epsilon)\sigma_0^2}\right) \tag{75}$$

$$= -\frac{1}{2}\log\left(1 - \frac{(1 - r)^2}{1 - r}\right) \tag{76}$$

$$= -\frac{1}{2}\log\left(\frac{\epsilon}{\sigma_0^2}\right). \tag{77}$$

To summarize, we obtained

$$D = \frac{1}{2}\mathbb{E}\left[(X_0 - X_1)^2\right] = \frac{1}{2}\epsilon, \quad R = \mathcal{I}(X_0; X_1) = -\frac{1}{2}\log\left(\frac{\epsilon}{\sigma_0^2}\right), \tag{78}$$

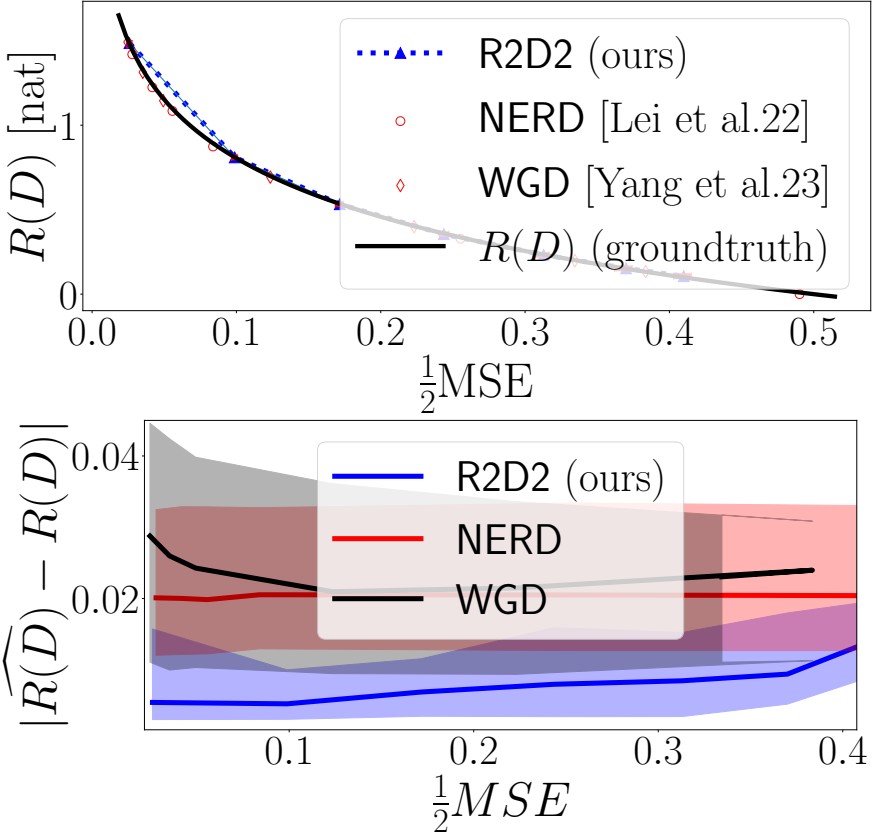

Figure 7: **The $R(D)$ function of a 1-D Gaussian source. (top)** Here, $X_0 \sim \mathcal{N}(0,1)$ and we applied Alg. 1 to $\epsilon \in [0.05, 0.95]$. *Green* markers indicate higher precision. We also plot the analytical result (*black* line). **(bottom)** We compare our algorithm to NERD and WGD, where we observe that our method is more accurate.

and recovered the well-known result

$$R_{\text{Gauss}}(D) = \frac{1}{2} \log \left( \frac{\sigma_0^2}{2D} \right), \quad 0 \leq 2D \leq \sigma_0^2. \tag{79}$$

Note that the factor of 2 is due to our convention $D = \frac{1}{2} MSE$. This result can be easily generalized for Gaussian vector sources and $\epsilon$'s smaller than the eigenvalues of the covariance matrix $\Sigma$. In this case, the solution to Eq. (17, BHE) is

$$p_t(x) = \mathcal{N}(0, \Sigma - \epsilon t I). \tag{80}$$

In Fig. 7 we demonstrate the efficiency of Alg. 1 on the 1-D case. We compare our method with NERD (Lei et al., 2022) WGD (Yang et al., 2024), where we observe that R2D2 is clearly superior to the existing methods in terms of estimation error, in both the high-rate and low-rate regimes.

### D.2 MIXTURE OF GAUSSIANS

The latter Example can be easily extended to the case of a Gaussian mixture, where

$$p_0(x) = \sum_{i=1}^{N} \frac{p_i}{\sqrt{2\pi\sigma_i^2}} e^{-\frac{(x-\mu_i)^2}{2\sigma_i^2}}. \tag{81}$$

As Eq. (17, BHE) is a linear equation, the solution is now given by the superposition

$$p_t(x) = \sum_{i=1}^{N} \frac{p_i}{\sqrt{2\pi(\sigma_i^2 - \epsilon t)}} e^{-\frac{(x-\mu_i)^2}{2(\sigma_i^2 - \epsilon t)}}, \quad \epsilon \in \left[0, \min_i \sigma_i^2 \right). \tag{82}$$

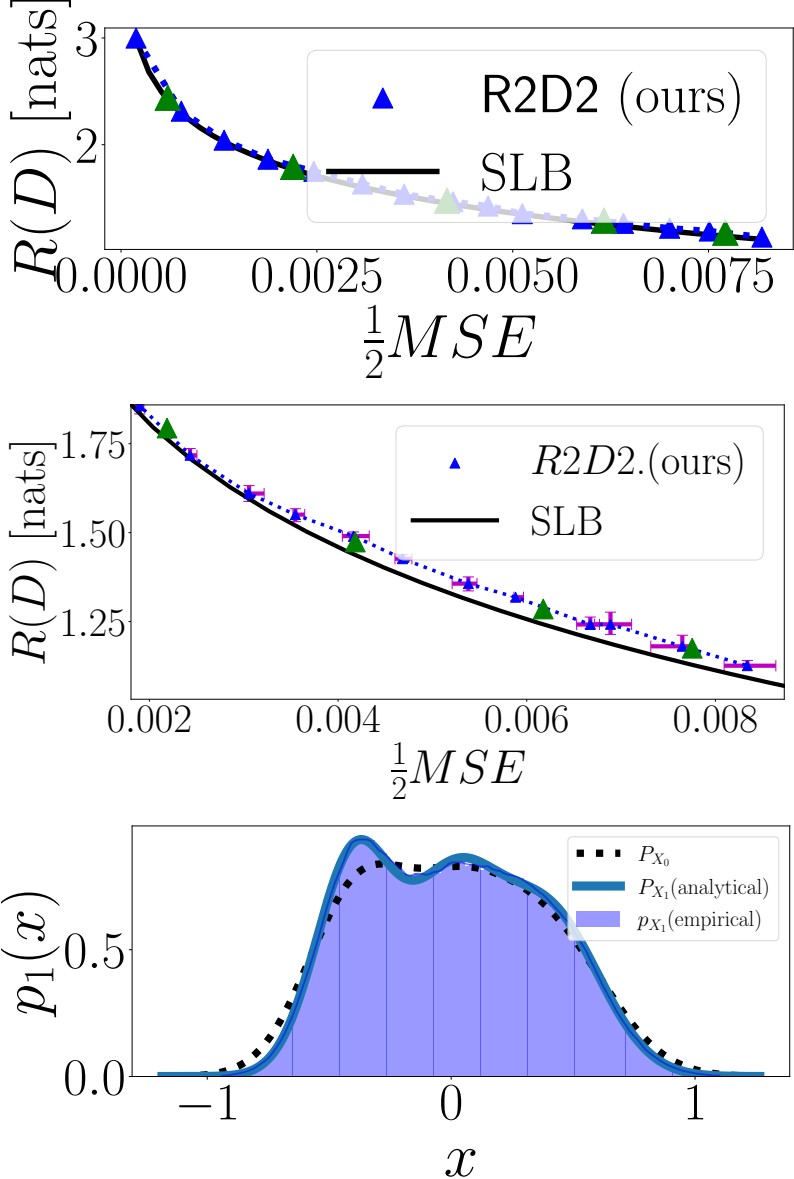

Figure 8: **The $R(D)$ function for a mixture of Gaussians.** Here, $X_0 \sim \mathbb{P}_0$ is a mixture of Gaussians where $\mu_i = -.4, 0, .4$, $\sigma_i^2 = 4 \times 10^{-2}, 5 \times 10^{-2}, 6 \times 10^{-2}$, $p_i = \frac{1}{3}$. (**top**) We apply Alg. 1 to $\epsilon \in [4 \times 10^{-4}, 1.64 \times 10^{-2}]$ and compare the result with SLB. *Green* markers indicate higher precision. (**middle**) A closer look on error bars (inter-quartile range) over 8 evaluations. (**bottom**) For $\epsilon = 1.56 \times 10^{-2}$, we plot the reconstruction distribution $\mathbb{P}_1$ which is the distribution of the diffusion process' outcome, $X_1$. Observe that the empirical distribution matches the analytical result (*bold* line).

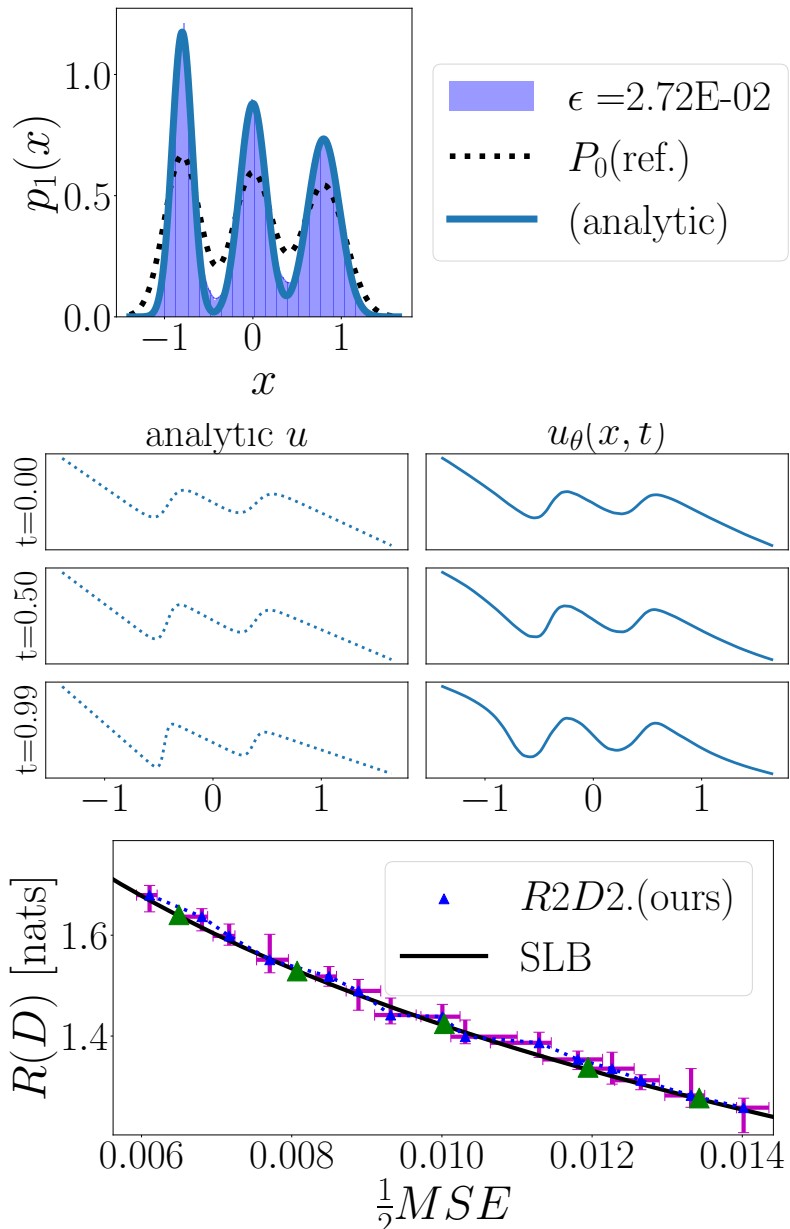

Figure 9: **The $R(D)$ function for a mixture of Gaussians (second example).** Here, $X_0$ is a mixture of Gaussians where $\mu_i = -.8, 0, .8$, $\sigma_i^2 = 4 \times 10^{-2}, 5 \times 10^{-2}, 6 \times 10^{-2}$, $p_i = \frac{1}{3}$. (**top**) The empirical reconstruction distribution for $\epsilon = 2.72 \times 10^{-2}$ matches the analytical result. (**middle**) Trained controller model $u_\theta$ (for $\epsilon = 2.8 \times 10^{-2}$), compared to analytical result at times $t = 0, 0.5, 0.99$. (**bottom**) RD function estimated over 16 evaluation steps (medians and inter-quartile ranges).

The optimal controller $u(x,t) = \epsilon \nabla \log p_t(x)$ is derived accordingly. We illustrate this result in Fig. 8, where $N = 3$ and $\mu_i = -.4, 0, .4$, $\sigma_i^2 = 4 \times 10^{-2}, 5 \times 10^{-2}, 6 \times 10^{-2}$, $p_i = \frac{1}{3}$. We apply Alg. 1 to $\epsilon \in [4 \times 10^{-4}, 1.64 \times 10^{-2}]$ and compare the estimated RD function with Shannon's lower bound (SLB)(Cover and Thomas, 2012; Berger, 2003)

$$H(\mathbb{P}_0) - \frac{1}{2}\log(4\pi eD), \tag{83}$$

approximated here from $M = 2^{11}$ i.i.d. samples $X^m \sim \mathbb{P}_0$ by

$$H(\mathbb{P}_0) \approx -\frac{1}{M}\sum_{m=1}^{M} \log p_0(X^m). \tag{84}$$

In practice, we estimated Eq. (84) for 8 independent trials, and used the median value for our approximation. For $\epsilon = 1.56 \times 10^{-2}$. We further plot the reconstruction distribution $\mathbb{P}_1$, which is the probability law of the diffusion process' outcome. We observe that empirical distribution obtained by Alg. 1 matches the closed-form result Eq. (82).

In the setting of Gaussian mixtures, we conducted an additional experiment in which $\mu_i = -.8, 0, .8$. The results are given in Fig. 9, where we also plot the outcome of the deep controller model we train, compared to the desired product.

### D.3 NON-GAUSSIAN MIXTURE

Consider the source $X_0$ drawn from the mixture $p_0(x) = \sum_{i=1}^{N} p_i C_i^{-1} sinc^4(\frac{x}{m_i})$ where $sinc(x) \triangleq \frac{\sin(x)}{x} \in \mathcal{C}^\infty(\mathbb{R})$, and $C_i = \frac{2\pi}{3}m_i$ are appropriate normalization factors. Recall that the characteristic function in this case is

$$\hat{p}_0(\omega) = \int_{-\infty}^{\infty} p_0(t)e^{-iwt}\mathrm{d}t = \sum_{i=1}^{N} p_i C_i^{-1} m_i \tilde{p}_0(m_i\omega), \tag{85}$$

where (with $*$ being the *convolution* operation)

$$\tilde{p}_0(\omega) = \tag{86}$$

$$= \frac{1}{2\pi}\left[\pi(1 - \frac{1}{2}|\omega|)_+ * \pi(1 - \frac{1}{2}|\omega|)_+\right] \tag{87}$$

$$= \frac{1}{96}\pi\left((w-4)^3\,\mathrm{sign}\,(w-4) - 4(w-2)^3\,\mathrm{sign}\,(w-2)\right. \tag{88}$$

$$\left. +6w^3\,\mathrm{sign}\,w - 4(2+w)^3\,\mathrm{sign}\,(2+w) + (4+w)^3\,\mathrm{sign}\,(4+w)\right), \tag{89}$$

which vanishes outside $\{|w| \leq 4\}$. Now, for non-vanishing mix distributions and small enough $\epsilon$'s, by Eq. (19) we can write the solution to Eq. (17, BHE) as

$$p_t(x) = \frac{1}{2\pi}\int_{-4}^{4} e^{i\omega x + \frac{1}{2}\epsilon\omega^2 t}\hat{p}_0(\omega)\mathrm{d}\omega = \frac{2}{2\pi}\sum_{i=1}^{N} p_i C_i^{-1} m_i \int_0^4 \tilde{p}_0(m_i\omega)\cos(\omega x)e^{\frac{1}{2}\epsilon\omega^2 t}\mathrm{d}\omega. \tag{90}$$

We numerically approximate

$$p_1(x) \approx \frac{1}{\pi}\sum_{i=1}^{N} p_i C_i^{-1} m_i \sum_{k=0}^{K-1} \frac{4}{K}\tilde{p}_0(m_i\frac{4k}{K})\cos(\frac{4k}{K}x)e^{\frac{1}{2}\epsilon(\frac{4k}{K})^2}. \tag{91}$$

Figure 10 demonstrates this result for $N = 4, m_i = \left[1, \sqrt{2}, \pi, e\right]$ ($m_i$'s were chosen such that $p_0(x) > 0$ everywhere on $\mathbb{R}$) and $p_i = \frac{1}{4}$, where we numerically integrated Eq. (91) with $K = 4 \times 32 \times 10^4$ in order to approximate the reconstruction distribution $p_1(x)$ for different values of $\epsilon$. We emphasize that although this is a toy problem, to the best of our knowledge, no other technique is known to tackle this case.

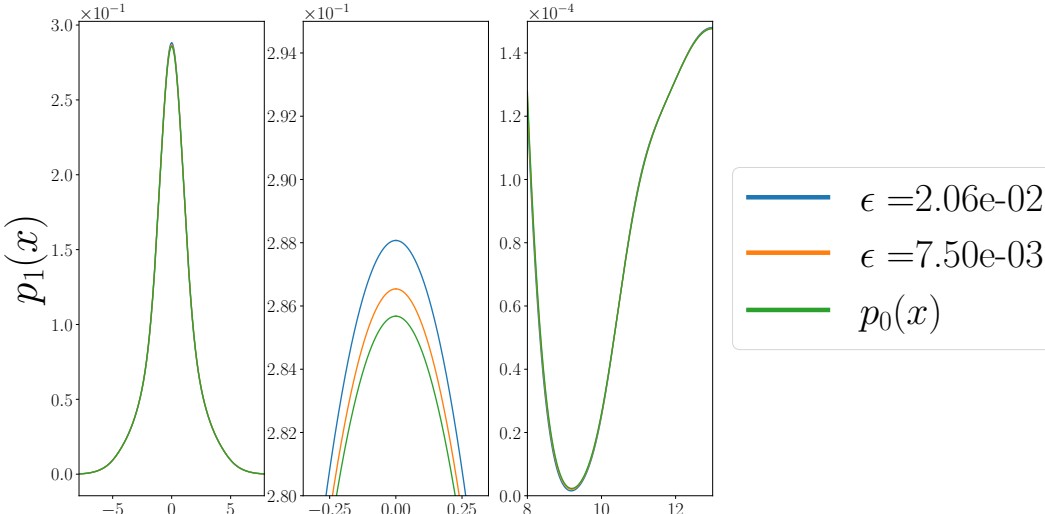

Figure 10: **Reconstruction distribution** $p_1(x)$ **of non-Gaussian mix source.** In the **(left)** pane, we approximate reconstruction distribution Eq. (90) by numerical integration. A closer look at different points is provided in the **(middle)** and **(right)** panes.

## D.4 EXAMPLE: CIFAR10 DATASET

We now demonstrate the efficiency of Alg. 1 on a realistic high-dimensional source. More specifically, as input to the Algorithm, we sample $4 \times 4$ grayscale image patches from the 'Cifar10' dataset (Krizhevsky and Hinton, 2009). Pixel values are normalized to $[0, 1]$. Fig. 11 demonstrates the efficiency of our method in solving this problem. In the upper pane, we present the RD function, as estimated by R2D2 (Alg. 1). In the lower pane, we present images, drawn from the reconstruction distribution $\mathbb{P}_1$ for different $\epsilon$'s.

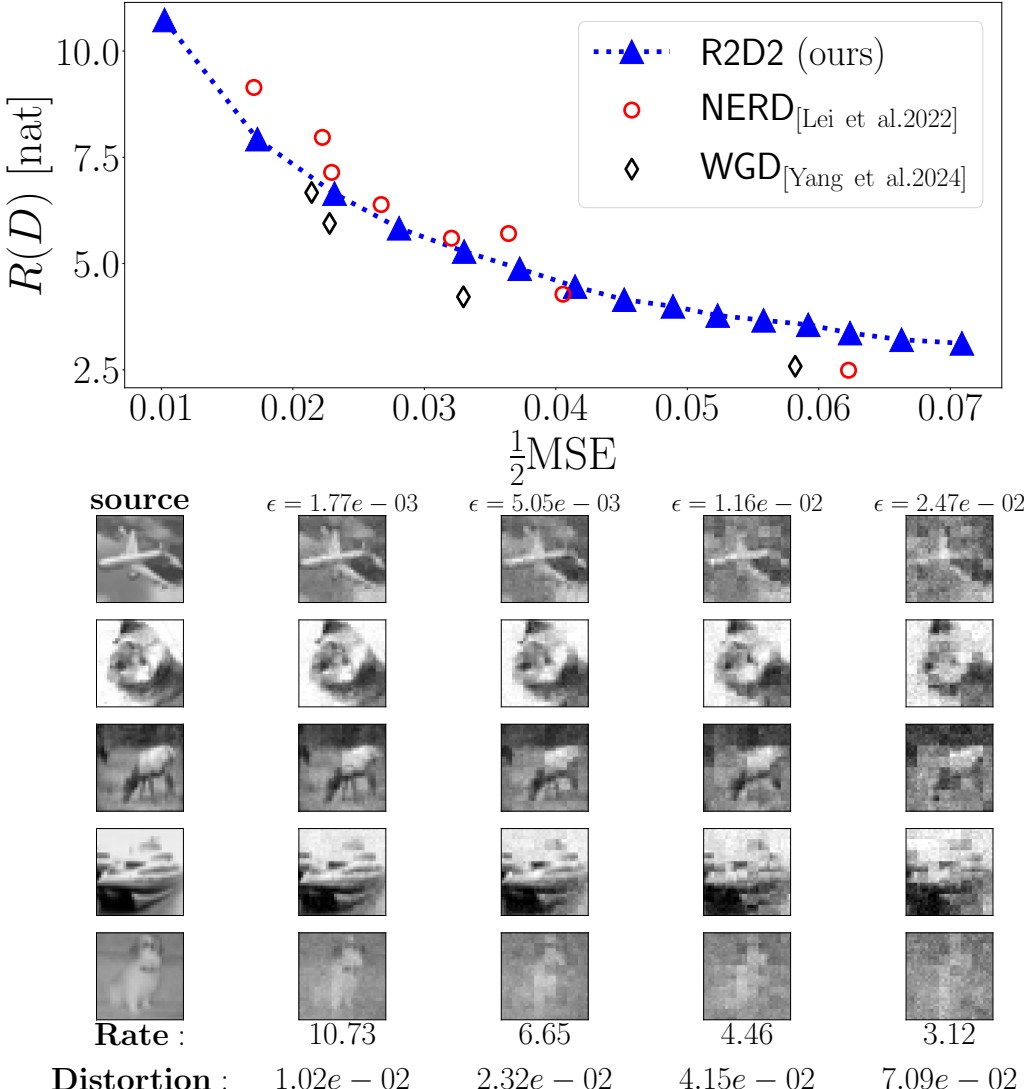

Figure 11: **The** $R(D)$ **function of Cifar10 images** (best viewed on screen). Here, $X_0$ is a random $4 \times 4$ patch from a grayscale image. **(top)** The RD function, estimated by R2D2 (Alg. 1). **(bottom)** Patches drawn from the reconstruction distribution $X_1$ obtained for different $\epsilon$'s.

# E   IMPLEMENTATION NOTES

## E.1   GENERAL DETAILS

For all experiments, we used 2 fully-connected DNN models:

- The controller $u_\theta$ taking $(X_t, t, \epsilon) \in \mathbb{R}^d \times [0, 1] \times [\epsilon_{\min}, \epsilon_{\max}]$ as input and whose output is in $\mathbb{R}^d$.
- In 1-D experiments we used $Z_\omega$ network for estimating the negentropy (in 1-D experiments), taking $(z, \epsilon) \in \mathbb{R}^d \times [\epsilon_{\min}, \epsilon_{\max}]$ and returning a scalar value (see App. B for details).
- For real dataset experiments, we alternatively used a network for representing KNIFE parameters.

Despite having different input and output layers, both models are of the same depth and hidden-layer sizes. We used LeakyReLU activation following each hidden layer. The two models were trained in a $1 : 3$-ratio of update steps, using the ADAM optimizer (Kingma, 2014) with parameters $\beta = (.9, .999)$ and learning rate $\alpha$.

At each evaluation step, we draw a batch of samples and evaluate $R(D)$ according to Alg. 1. Whenever there are more than one evaluation step or independent seeds, the presented $(R, D)$-values are the medians over all steps, while error bars indicate the inter-quartile (25%-75%) range.

Codes for the NERD baseline (Lei et al., 2022) are provided by the authors at https://github.com/leieric/NERD-RCC. Codes for the WGD baseline (Yang et al., 2024) are provided by the authors at https://github.com/yiboyang/wgd. Our codes are publicly available at https://github.com/ML-group-il/r2d2.

All experiments were implemented in PyTorch (Paszke, 2019) environment, and performed using a NVIDIA RTX A6000 GPU.

## E.2   SIMULATION PARAMETERS

**Gaussian sources (Fig. 2)**

- $\epsilon_{\min}, \epsilon_{\max} = 0.025, 0.975$
- DNN hidden layers: 1
- Hidden layer size: 128
- Step size: $\Delta_t = \frac{1}{100}$
- Train steps: 25,000
- Batch size (train): $M = 512$
- Experiments: 64
- Batch size (evaluation): 1024
- Learning rate: $\alpha = 1e\text{-}3$

**Mixture of Gaussians (Fig. 4)**

- $\epsilon_{\min}, \epsilon_{\max} = 4e\text{-}4, 1.64e\text{-}2$
- DNN hidden layers: 4
- Hidden layer size: 128
- Step size: $\Delta_t = \frac{1}{100}$
- Train steps: 3.7M
- Batch size (train): $M = 256$
- Evaluation steps: 8 (32 at high-precision)
- Batch size (evaluation): 1024
- Learning rate: $\alpha = 5e\text{-}4$

**Mixture of Gaussians** $#2$ **(Fig. 9)**

- $\epsilon_{\min}, \epsilon_{\max}$ = 1.2e-2,2.8e-2
- DNN hidden layers: 4
- Hidden layer size: 100
- Step size: $\Delta_t = \frac{1}{100}$
- Train steps: 855,000
- Batch size (train): $M = 256$
- Evaluation steps: 16 (64 at high-precision)
- Batch size (evaluation): 1024
- Learning rate: $\alpha = $ 1e-3

**Cifar10 dataset (Fig. 5)**

- $\epsilon_{\min}, \epsilon_{\max}$ = 0.0005,0.026
- DNN hidden layers: 3
- Hidden layer size: 1024
- Step size: $\Delta_t = \frac{1}{128}$
- Train steps: 292,968
- Batch size (train): $M = 256$
- Evaluation steps: 128
- Batch size (evaluation): 2048
- Learning rate: $\alpha = $ 2.5e-4
- KNIFE parameter: $K = 512$

