# OpenReview forum: "An Optimal Diffusion Approach to Quadratic Rate-Distortion Problems: New Solution and Approximation Methods"
_ICLR.cc/2026/Conference — ICLR 2026 Poster_

### Official Review · Reviewer_ndvv · 2025-10-21

**Soundness:** 4
**Presentation:** 3
**Contribution:** 2
**Rating:** 8
**Confidence:** 4

**Summary:**

This paper introduces a method to compute the _rate-distortion function_ $R(D)$, which is the smallest information rate needed to achieve a given level of distortion $D$. The approach focuses on the case where the data source is continuous, and the average distortion is quadratic, $D(X, \hat{X})=\tfrac{1}{2} \mathbb{E}[\| X - \hat{X} \|^2]$. Then, the problem reduces to an entropic optimal transport problem, which, in the low distortion regime, can be mapped to a Schrödinger bridge problem. Through this chain of reasoning the task of determining $R(D)$ is converted to a free energy minimization problem, which the authors refer to as _terminal-entropy stochastic control_, wherein the optimal rate corresponds to a balance between control energy and terminal state uncertainty. This optimum can be determined by parameterizing the control with a neural network and scanning through the free-energy landscape, which R2D2 algorithm presented in the paper. Experiments with some mixture distributions and the CIFAR-10 dataset demonstrate the algorithm at work.

**Strengths:**

1. The paper leverages both established literature and new developments in stochastic control theory and entropic optimal transport to address their central problem.
2. The proposed solution is elegant, and the argument is generally well-presented.

**Weaknesses:**

1. The authors state that their approach is restricted to quadratic distortion, and the low-distortion regime. Why is this corner interesting from a practical standpoint? Or was it chosen because it makes the problem tractable? The paper would benefit from motivating this choice early on.

2. In its present state, the paper reads like a stochastic control paper which shows that the rate-distortion function is the minimum of a free energy functional. The numerical solution to the latter involves a neural network. Beyond this, the paper makes no wider connection to machine learning in general. See my second question below.

3. Per my understanding, the method does not scale well to higher dimensions. The main bottleneck is the estimation of $H(X_1)$, as explained in appendix B of the paper. If this is an actual limitation it should be highlighted earlier in the paper. See my third question below.

4. Not really a weakness, but the derivations in Sec. 3.2 appears to be mostly a special case of the one presented in [1]. Include this reference.

Minor comment about formatting: The captions for the subfigures in Figure 1 could be spaced apart more for readability.
Minor comment about references: I encourage you to cite [4], which is an English translation of Schrodinger's paper 'On the reversal of the laws of nature.'

**Questions:**

1. The core arguments of the paper bear a strong resemblance to the _adjoint-matching_ paper from last year [2]. Briefly, given samples of a distribution $p_{\rm base}$ that paper addresses the problem of sampling from the tilted distribution $e^{r(x)} p_{\rm base}$. It seems to me that this is also a form of distortion. While $r(x)$ is fixed in their case, making it free may allow their argument to be extended to some class of distortions besides the one addressed in your paper. Could this be an interesting direction for follow-up work?

2. The terminal entropy stochastic control picture in Sec. 5 looks a lot like max-entropy RL, which you mention in the introduction. Both problems extremize an expected cost (or distortion) while regularizing by an entropy. Does this speak to a deeper connection between compression and learning? Is it correct to say that _the optimal policy is the most efficient compressor of the admissible solutions at a fixed cost_? Sharpening such connections could make the paper much more compelling.

3. On the difficulty of determining $H(X_1)$ in high-dimensions, if you are simulating the trajectories for different $u_\theta$ in Algorithm 1, is it possible to use the samples at $t=1$ to compute that entropy, say with the approach from [3] you cited?

4. Around line 207, what does it mean when you write $W_0^\epsilon \sim \mathbb{P}_0$? If $\mathbb{P}_0$ denotes the source distribution, is $W_0^\epsilon$ the initial state under a small diffusion of it?

Overall I think this is a good paper that can be further improved with some minor changes. I would also appreciate if the authors can address my comments/questions.

[1] Pavon, M., “Stochastic control and nonequilibrium thermodynamical systems,” Applied Mathematics & Optimization, vol. 19, pp. 187-202, Jan. 1989.

[2] Domingo-Enrich, C., Drozdzal, M., Karrer, B. & Chen, R. T. Q. "Adjoint Matching: Fine-tuning Flow and Diffusion Generative Models with Memoryless Stochastic Optimal Control," ICLR 2025.

[3] Franzese et al., “MINDE: Mutual Information Neural Diffusion Estimation,” ICLR 2024

[4] Chetrite, Raphaël; Muratore-Ginanneschi, Paolo; Schwieger, Kay. “E. Schrödinger’s 1931 paper ‘On the Reversal of the Laws of Nature’ [‘Über die Umkehrung der Naturgesetze’, Sitzungsberichte der preussischen Akademie der Wissenschaften, physikalisch mathematische Klasse, 8 Nº 9 144-153].” arXiv preprint arXiv:2105.12617 (2021).

---

> ### Author Response · Authors · 2025-11-27
> **Response to Reviewer ndvv (1/2)**
>
> Thank you very much for your positive review, helpful comments and great pointers. As you suggested, our paper can be further improved with minor additional efforts and we will gladly implement any additional idea.
>
> Regarding your concerns:
>
> ---
>
> 1. **Motivate the choice of MSE loss and low-distortion regime.**
>
> A: Although looks confining, the MSE loss is a central quantity of interest in RD theory [Shannon, 1959; Berger,2003]. This is especially true in the low-distortion regime, since on a finite-dimensional data all distances are equivalent (up to a constant), so small MSE indicates similarity to the reconstructed letter.
> The low-distortion regime is of a special interest too, since in modern communication high bandwidth channels are often available, so data should be compressed using higher bit rates with small errors [Chafii et al., 2023].
>
> We added this remark to the revised paper.
>
> [Marwa Chafii, Lina Bariah, Sami Muhaidat, and Merouane Debbah. Twelve scientific challenges for 6g: Re-
> thinking the foundations of communications theory. IEEE Communications Surveys & Tutorials, 25(2):868–904,
> 2023.]
>
> ---
>
> 2. **Wider connection to machine learning in general: Is there a deeper connection between compression and learning? Connection to maxEntRL? Can optimal policy be viewed as a compressor? such connections could make the paper more compelling.**
>
> A: Information theory is a foundational field, bearing a strong impact across science and engineering. Since Rate-Distortion is a core problem within information theory, which has been extensively  studied for almost seven decades, we expect our newly proposed perspective and methods to be appealing to a broad community.
>
> The motivation we kept in mind during the work was indeed thinking of the controller as emulating the encoding-decoding action. However, rigorously speaking, there is no really an 'encoder' *nor* a 'decoder' here, but rather a generative model that cast the source distribution to its final distribution (reconstruction law in this case), where the properties of the model reveal some properties of the solution to RD. To give a hint of this motivation, we added the following phrase to the intro:
> ```
> " It then becomes natural to investigate diffusion processes in the context of RD theory, as generative models casting the source probability to the distortion-optimal reconstruction distribution. As we show in this work, this fresh point of view reveals surprising analytical results as well as novel estimation methods. "
> ```
>
> Beyond this motivation, note that as we formulate RD as stochastic control, the latter generally possesses a close connection to estimation and learning through the celebrated Kalman `duality' [Todorov, 2008], but this tends to take nontrivial forms for non linear/quadratic problems. This can be an interesting idea for future work.
>
> [Todorov, Emanuel. "General duality between optimal control and estimation." 2008 47th IEEE conference on decision and control. IEEE, 2008.]
>
> ---
>
> 3. **Entropy estimation does not scale well to high dimensions.  This should be highlighted.**
>
> A: We will highlight this point in the revision. We note, however that while the entropy estimator is an important component in Algorithm 1, the entropy estimation problem itself is a technical step, orthogonal to our findings. Any future improvement in entropy estimation would directly lead to an improvement of our RD estimator.
>
> ---
>
> 4. **Derivations in Sec. 3.2 are similar to the ones presented in [1].**
>
> A: In the revision we discuss [1] and the connection to our work in detail, in the related work section:
> ```
> " After submitting this manuscript for publication, we became aware of the work of Pavon (1989), which, within the context of physical systems, derived a variational form similar to (15,varTEC). There, the controlled state and observation have pointwise initial conditions and control is reversed in time. To the best of our knowledge, our work is the first to present such a formalism in the context of information theory... "
> ```

---

> ### Author Response · Authors · 2025-11-27
> **Response to Reviewer ndvv (2/2)**
>
> 5. **The core arguments of the paper bear a strong resemblance to
> Fine-tuning with adjoint-matching. Could this be an interesting direction for follow-up work?**
>
> A: Adjoint matching is a method for accelerating inference in diffusion models, stochastic control and SB [Liu et al. 2025]. Accelerating our proposed method using adjoint matching is indeed a subject for future work. However,  the current definition of adjoint ( eq. (29)-(31) in Domingo-Enrich et al. (2025) ) should be then generalized to fit our setting (since the "terminal" entropy cost in our work is a *functional* of $\mathbb{P}_1^u$, rather than a function of $x_1$).
>
> The use of `tilted' distributions to match some distortion is also an interesting idea.  We should pay attention that:
>
> **I)** The 'distortion' measure $r(x_1)$ is a non-reference one (independent of the source $x_0$). Such measures can describe `likelihood' under some reference probability (*e.g.*, how likely the outcome is a natural image).
>
> **II)** In order to admit a `tilted' probability as described, the base process should be designed to be **memoryless**. This will also change the underlying distortion (not MSE) such that (see [Gushchin, et al., 2022] App.I)
> $$d(x_0,x_1) = -\log p_{base}(x_1|x_0)=  -\log p_{base}(x_1).$$
>
> To summarize, as future work, it is possible to look at a problem of minimizing a tradeoff between non-reference distortion measures and cross-entropy between $p_{x_1}$ and a guidance measure $r(x) \propto -\log p_{guide}(x)$.
>
> [Liu, Guan-Horng, et al. "Adjoint Schr\" odinger Bridge Sampler." arXiv preprint arXiv:2506.22565 (2025)]
>
> [Nikita Gushchin, Alexander Kolesov, Alexander Korotin, Dmitry Vetrov, and Evgeny Burnaev. Entropic
> neural optimal transport via diffusion processes. arXiv preprint arXiv:2211.01156, 2022.]
>
> ---
>
> 6. **Is it possible to use $X_1$ samples to compute the entropy?**
>
> A: We indeed use the edge samples at time $T=1$ to compute the entropy, as described in Alg. 1 and App. B. We have also tried various recent techniques such as MINDE [Franzese et al, 2023] and InfoBridge [Kholkin et al., 2025], but those turned out to get unstable during the training phase.
>
> [Giulio Franzese, Mustapha Bounoua, and Pietro Michiardi. MINDE: Mutual information neural
> diffusion estimation. arXiv preprint arXiv:2310.09031, 2023.]
> [Sergei Kholkin, Ivan Butakov, Evgeny Burnaev, Nikita Gushchin, and Alexander Korotin. Info-
> Bridge: Mutual information estimation via bridge matching. arXiv preprint arXiv:2502.01383,
> 2025.]
>
> ---
>
> 7. **What is the process $W^\epsilon_t$?**
>
> The `base' process $W^\epsilon_t$ is similar to the standard Wiener process (Brownian motion) whose variance is scaled by $\epsilon$, except its **initial state** $W^\epsilon_0$ has the same distribution as the source $\mathbb{P_0}$ (while the standard Wiener process is defined to start at $W_0=0$ a.s.). This is written $W^\epsilon_0 \sim P_0$. We apparently missed the subscript $t$ in l.207, and we fixed this typo in the revision.
>
> ---
>
> **Minor comments:**
>
> * We formatted the captions of  Figs. 1 and 2  as you suggested.
>
> * Thank you for pointing out the English translation of
> [4], which is a great pointer for us and our (non-French) readers!

---

### Official Review · Reviewer_GSoo · 2025-10-27

**Soundness:** 3
**Presentation:** 4
**Contribution:** 3
**Rating:** 8
**Confidence:** 3

**Summary:**

This paper exploits the connection between rate-distortion (RD) and entropic optimal transport to propose a novel stochastic control formulation, and use a classic result dating back to Schrodinger to show that the tradeoff between rate and mean squared error distortion is equivalent to a tradeoff between control energy and the differential entropy of the terminal state, whose probability law yields the reconstruction distribution. For a special class of sources, they show that the optimal control law and trajectory in the space of probability measures are given by solving a backward heat equation. In the more general case, their approach gives rise to a numerical solution method, estimating the RD function using diffusion processes with a constant diffusion coefficient.

**Strengths:**

They present a novel stochastic control formulation that is regularized by terminal uncertainty, and show that this formulation is equivalent to the RD problem.
They characterize the optimal solution under some regularity conditions.
Found a closed-form solution for the reconstruction distribution of a Gaussian-mixture source.
Proposed a novel neural method for estimating the RD function and the reconstruction distributions, using a simple diffusion model.

**Weaknesses:**

In some figures, the font size look weird. This is in the Appendix.

**Questions:**

Section 3.3 seems to focus on mixtures. Will there be any non-mixture examples?

---

> ### Author Response · Authors · 2025-11-27
> **Response to Reviewer GSoo**
>
> Thank you very much for your positive review. We will gladly address any other comment.
>
> Regarding your concerns:
>
> ---
>
> 1. **In some figures, the font size look weird. This is in the Appendix.**
>
> A: Thank you for taking a careful look at our paper and Appendix. We have used large fonts for the ease of reading. If this is not the case, please let us know in what figures we can improve.
>
> ---
>
> 2. **Are there any non-mixture examples to Thm.3.2?**
>
> A: Thank you for this question. Thm 3.2 results can be applied to strictly positive and smooth probabilities. We used the sinc-mixture example since it is band limited and Fourier analysis could be used. Other possible non mixture examples where a similar frequency domain analysis could be carried are of a uniform or a normalized $sinc^4$-distributed signal with an additive Gaussian noise.

---

### Official Review · Reviewer_GzUT · 2025-11-01

**Soundness:** 2
**Presentation:** 2
**Contribution:** 2
**Rating:** 2
**Confidence:** 3

**Summary:**

The authors consider the problem of probabilistic data compression while keeping the distortion rate at an acceptable level. They look at this problem through the lens of entropy-regularized stochastic optimal control and suggest the Terminal-Entropy Control (TEC) method aiming at minimizing the loss function with respect to the admissible control $u(x, t)$ and the terminal distribution $\mathbb P_1$. The authors elaborate on connections of the TEC formulation with the approach based on entropic optimal transport (Theorem 3.1). They also derive an explicit expression for the optimal control function $u^\star(x, t)$ (Theorem 3.2). Finally, the authors illustrate the performance of their procedure with numerical experiments on artificial and real-world data.

**Strengths:**

According to the numerical experiments, the method outperforms its competitors.

**Weaknesses:**

1. I do not see a reason why numerical experiments on one-dimensional Gaussian data and CIFAR-10 were not attached as a supplement. This raises concerns about reproducibility of the results reported in Section 5.

2. It seems that the authors missed a very relevant paper [Dai Pra, 1991], where the author studied the problem of stochastic optimal control. His findings were recently applied to generative modelling (see, for instance, [Rapakoulias et al., 2023] and [Puchkin et al., 2025]). I suppose that the proof of Theorem 3.2 in the present submission can be simplified if the authors took into account Theorem 3.2 from [Dai Pra, 1991], where the author derived an explicit expression of the optimal control function $u^\star(x, t)$ for a fixed $\mathbb P_1$ through Schrodinger potentials (see, e.g., [Korotin et al., 2024] for the definition). I would be grateful if the authors could elaborate on this point.

3. In the suggested algorithm, one has to optimize both the terminal distribution $\mathbb P_1$ and the control $u(x, t)$. However, for any fixed $\mathbb P_1$ the form of the optimal control $u^\star(x, t)$ is known (see Theorem 3.2 in [Dai Pra, 1991]). Moreover, Dai Pra [1991] derives the corresponding value of $(2\varepsilon)^{-1} \mathbb E \int_0^1 \|u^*(x, t)\|^2 \, dt$. Hence, the loss can be substantially simplified.

In view of the weaknesses 2 and 3, I would not recommend the paper for acceptance in its present form. I think that it will benefit from a revision, if the authors take into account the results of Dai Pra.


**Minor remark**

On page 4 the authors write: ``Recent developments (Gushchin et al., 2022) has drawn an equivalency (up to an additive constant, depends on $\mathbb P_0$ , $\mathbb P_1$ , $\varepsilon$) between SB and EOT, where the latter can be optimized via a game-theoretic formulation.'' The connection between SB and EOT was known far before (Gushchin et al., 2022). In particular, it was mentioned in the survey of Leonard (2013).

**References**

[Dai Pra, 1991] Paolo Dai Pra. A stochastic control approach to reciprocal diffusion processes. Applied Mathematics
and Optimization, 23(1):313–329, 1991.

[Korotin et al., 2024] A. Korotin, N. Gushchin, and E. Burnaev. Light Schrodinger bridge. In The Twelfth
International Conference on Learning Representations, 2024.

[Puchkin et al., 2025] N. Puchkin, I. Pustovalov, Y. Sapronov, D. Suchkov, A. Naumov, and D. Belomestny. Sample complexity of Schrodinger potential estimation. Preprint. ArXiv:2506.03043, 2025.

[Rapakoulias et al., 2024] G. Rapakoulias, A. R. Pedram, and P. Tsiotras. Go With the Flow: Fast Diffusion for Gaussian Mixture Models. Preprint. ArXiv:2412.09059v3, 2024.

**Questions:**

1. Can you simplify the proof of Theorem 3.2 in the present submission using Theorem 3.2 from [Dai Pra, 1991]?

2. Can you simplify the loss (12) using Theorem 3.2 from [Dai Pra, 1991]? Is it possible to exclude $u$ from the optimization problem?

---

> ### Author Response · Authors · 2025-11-27
> **Response to Reviewer GzUT (1/2)**
>
> Thank you very much for your insightful review.
>
> If our comments resolve your concerns, please reconsider your score. We will gladly address any remaining issue.
>
> ---
>
> regarding your concerns:
>
> ---
>
> 1.**Attach code as supplemental**:
>
> A: The current submission platform is open to the public, which is great for clarity and for honest discussion. The downside of this feature, however, is a potential misuse of released assets by anonymous viewers, hence we didn't include our code with the submission.
> We are very sorry if this situation is somewhat inconvenient.
>
> We have arranged our experiment codes, and will release them with our upcoming revisions.
>
> ---
>
> 2. **Missed references**:
>
> A: Thank you very much for pointing out the very relevant papers of [Dai Pra, 1991] [Puchkin et al., 2025] [Rapakoulias et al., 2024]. We mention them in our revision.
>
> ---
>
> 3. **Dai Pra [1991] derives the control and losses according to $\mathbb{P}_1$, can you simplify proof of Theorem 3.2 accordingly**:
>
> A:
> The interesting result of Dai Pra [1991], which you have noted, is based on expressing the optimal control as $ u=\epsilon \nabla \log h(x,t)$,  where $h$ is a functional of a kernel $q$ (Gaussian/Heat kernel under our setting) and the terminal potential $\rho_T$ of a schrodinger system, where terminal distribution is *known*. Even in this case, this form is not explicitly provided.
>
> Note that our setting differs from the classical SB problem, since we do not assume we know the terminal density $p_1^* (x) $ (only its existence, by assumption A1). Now, a given value of $p_1(x)$ affects both potentials of the system $\rho_0,\rho_T$. The cost is given again in terms of these general measures. Applying these implicit identities to our optimization, though apparently possible, is not straightforward. On the other hand, our current approach finds that $u^* (x,t)=\epsilon \nabla \log p_t^* (x)$ without any special difficulties, so we believe it is the simplest way to put it down. In addition, our proof doesn't require theoretical background about Schrodinger systems and potentials which makes it self-contained and more readable for a broader ICLR audience.
>
> ---
>
> 4. **In the suggested algorithm, one has to optimize over both terminal distribution
> and the control:**
>
> A key feature of presenting RD as a control problem (14,TEC) is that $\mathbb{P}_1$ is *free* (as emphasized therein), and completely determined by the controller (and the given $\mathbb{P}_0$). Hence, optimization is performed here w.r.t. the function $u$ **only**.  This approach is very common when formulating and solving SB and related problems (see, e.g. recent [Gushchin et al., 2022]).
>
> [Nikita Gushchin, Alexander Kolesov, Alexander Korotin, Dmitry Vetrov, and Evgeny Burnaev. Entropic
> neural optimal transport via diffusion processes. arXiv preprint arXiv:2211.01156, 2022.]

---

> ### Author Response · Authors · 2025-11-27
> **Response to Reviewer GzUT (2/2)**
>
> 5. **Can you revise the algorithm, taking into account the results of Dai Pra. Can you simplify the loss (12)? Is it possible to exclude $u$ from the optimization problem?}**
>
> A: In this work, we focused our efforts in establishing the connection between RD and stochastic control (TEC). Since control problems are naturally formulated as minimization over admissible controls (re our answer to 4), we formulated the loss (12) as presented, where the problem (13) can be solved under the conditions of Thm 3.2. Our proposed method (R2D2) is a rather straightforward translation of this idea, served for cases where regularity conditions fail to hold, or source distribution is unknown. Note that given a controller $u$, sampling from the terminal distribution $\mathbb{P}_1$ is immediate, using the Euler-Maruyama scheme (Alg.2, App.A).
>
> As you cleverly suggested, it is apparently possible to equivalently design the problem to depend only on the distribution $\mathbb{P}_1$. Loss (12) should be then reformulated to depend on the potentials of the system, $\rho_0$ and $\rho_1$. These (generally un-normalized) measures are affected by the choice of $\mathbb{P}_0,\mathbb{P}_1$, and under the SB setting could be sampled using various techniques (as in [Puchkin et al., 2025] ). Thus, optimization can be done w.r.t. $\mathbb{P}_1$, while $u$ is determined accordingly. This approach is orthogonal to ours, and is not straightforwardly adapted to our half-open setting. it is **substantially different** from the approach we take in this work. It certainly deserves a separate discussion and could be a subject for a follow-up work (which, to the matter of fact, we are *already* conducting). We will mention this in the discussion.
>
> ---
>
> 6. **Work will further benefit from the results of Dai Pra.**
>
> A: We thank you again for pointing out the interesting and important work of Dai Pra. Their approach may lead to a line of follow-up works, but these will be substantially different from this work (re our answers above).
>
> As we show in the paper, our `Optimal Diffusion' (or stochastic control) approach to RD is intriguing by itself and leads to new and striking results, unknown for decades to information theorists. Taking this into account, we thus  kindly ask you to consider recommending our paper for acceptance.
>
> ---
>
> **Minor remark**
>
> * Thank you very much for this remark, we will add the relevant references in place.

---

> > ### Comment · Reviewer_GzUT · 2025-11-27
> >
> > **1.** The responses 3 and 4 did not address my concerns. Probably, the authors misunderstood my message. For this reason, I would like to clarify it.
> >
> > To solve the terminal-entropy stochastic control problem (14, TEC) (with free density $\mathbb p_1$), the authors can fix $\mathbb P_1$, find the corresponding optimal control $u_{\mathbb p_1}^\star$, substitute it into the target functional, and then perform an optimization with respect to $\mathbb P_1$.
> >
> > For a fixed $\mathbb P_1$, the optimal control $u_{\mathbb P_1}^\star$ and the corresponding value of the target functional have analytic expressions (see Theorem 3.2 in [Dai Pra, 1991]). For this reason, there is no need to optimize the target functional (14, TEC) with respect to $u$. You can optimize it with respect to $\mathbb P_1$ (estimating the corresponding Schr\"odinger potential) and then find the corresponding optimal control $u_{\mathbb P_1}^\star$ using the analytic expression from [Dai Pra, 1991; Theorem 3.2]).
> >
> >
> > **2.** I am also confused by this phrase (point 3 of the response): "On the other hand, our current approach finds that $u^\star(x, t) = \epsilon \nabla \log p_t(x)$ without any special difficulties, so we believe it is the simplest way to put it down. In addition, our proof doesn't require theoretical background about Schrodinger systems and potentials which makes it self-contained and more readable for a broader ICLR audience."
> >
> > The result of Theorem 3.2 is a direct consequence of Theorem 3.2 from [Dai Pra, 1991] and the connection between Fokker-Planck equations and SDEs. Hence, its novelty is very limited. However, you present it as one of the contributions of the submission, ignoring relevant background. I doubt that it makes your paper more readable for a broader ICLR audience. However, I believe that it creates a wrong impression that there were no similar results in the literature.

---

> ### Author Response · Authors · 2025-12-03
> **Response to Official Comment by Reviewer GzUT**
>
> Thank you for the clarifications. We thoroughly studied the results of [Dai Par, 91] as you suggested, and believe we can now fully address your concerns.
>
> As we state in the manuscript, our work is focused on establishing a connection between RD theory and stochastic control. Since your concerns are mainly focused on the proof of our Thm.3.2, we hope that you recognize the overall novelty and importance of our work.
>
> Please find our detailed response below.
>
> ---
>
> 1.
>
> Q:
> ```
> To solve the terminal-entropy stochastic control problem (14, TEC) (with free density $p_1$), the authors can fix $p_1$, find the corresponding optimal control $u_{p_1}$, substitute it into the target functional, and then perform an optimization with respect to $p_1$.
> ```
> A:
> While we agree that this is indeed a valid approach to the problem (14, TEC), in our work we took the opposite approach, namely optimizing over $u$ and determining $p_1$ accordingly.
>
> Both approaches are equally sound, and can be found in the literature in the SB context, since they possess different advantages.
>
> While fixing $p_1$ as you suggest results in an analytical form of the controller's energy, the benefit of our approach is the absence of the need to evaluate the potentials of the Schrodinger system as we explain below. Note that our analytical result (17,BHE) eventually characterizes $p_1$ in terms of $p_0$ only, which is a parameter of the problem.
>
> Q:
> ```
> For a fixed $p_1$, the optimal control and the corresponding value of the target functional have analytic expressions (see Theorem 3.2 in [Dai Pra, 1991]). For this reason, there is no need to optimize the target functional (14, TEC) with respect to $u$. You can optimize it with respect to $p_1$ (estimating the corresponding Schrodinger potential) and then find the corresponding optimal control using the analytic expression from [Dai Pra, 1991; Theorem 3.2]).
> ```
>
> A:
> For a fixed $p_1$, the optimal control
> and the corresponding value of the target functional have analytic expressions, however these are given in terms of the system potentials $\rho_0,\rho_1$, rather than of $p_1$:
>
> $$
> u = \epsilon \nabla
> \int q(t,x,1,y)\rho_1(y)dy, \frac{1}{2\epsilon} \int_0^1 \|u\|^2 dt = D_{KL}(p_1, S_{1}\rho_0) - D_{KL}(p_0, \rho_0)
>  $$
>
> Now, our objective (14,TEC) will take the mixed form
>
> $$
> \frac{1}{2\epsilon}\int_0^1 \|u\|^2 dt+ H(p_1)= D_{KL}(p_1, S_1\rho_0) - D_{KL}(p_0, \rho_0) + H(p_1)
> $$
>
> As you noted, the potentials should be estimated, which is a not a trivial step numerically nor analytically, as they are generally given by an implicit form involving $\rho_0,\rho_1,p_1$,$q$ and $p_0$ (see [Dai Pra, 1991] Eq.3.13). As a consequence, optimizing (or solving analytically) this objective is not straightforward at all.
>
> On the other hand, optimizing over $u$ eliminates the dependence in evaluating or sampling the potentials.
>
>
> ---
>
> 2.
>
> Q:
> ```
> I am also confused by this phrase...
> ```
>
> A:
> What we meant to say is that our approach leads (under the conditions of Thm 3.2) to the simple expression $u^*(x,t) = \epsilon \nabla \log p_t(x)$, where $p_t$ is a trajectory determined by (17, BHE). This expression does not involve the potentials of SB, as we mentioned above, which makes it simpler to understand and utilize.
>
>
> Q:
> ```
> The result of Theorem 3.2 is a direct consequence of Theorem 3.2 from [Dai Pra, 1991].
> ```
>
> A:
> We beg to differ. While may share similar motivation with SB, Thm.3.2 is in no way a "direct consequence of Theorem 3.2 from [Dai Pra, 1991]". Our work discusses a different setting (TEC) from SB discussed in [Dai Pra].  Our motivation is different - solving RD problems through a control formulation. Moreover, the results of [Dai Pra] are given in implicit forms, involving the (unknown) Schrodinger potentials, which, as you noted, should be "estimated". This is not necessary in our approach.
>
> We insist that our contributions (including Thm.3.2 and its applications) are strong and novel. For evidence, they reveal new analytical properties of RD problems, unknown for decades for information-theorists.
>
>
> Q:
> ```
>  I believe that it creates a wrong impression that there were no similar results in the literature.
> ```
>
> A:
> We appreciate your concern, and agree that our paper should provide full and detailed background and existing knowledge to the benefit of our readers. In our revised paper, we made a substantial effort to mention all relevant results (under Related Work of Sec.2.2).
>
>
> ---
>
> As a final remark,  In your original review you wrote (W3):
> "In the suggested algorithm, one has to optimize both the terminal distribution $p_1$ and the control $u$."
> In our response 4 we clarify that our optimization is done only w.r.t. the control  $u$.  We thus do not understand how this fails to address your concern.

---

### Official Review · Reviewer_ZRmZ · 2025-11-01

**Soundness:** 3
**Presentation:** 3
**Contribution:** 3
**Rating:** 6
**Confidence:** 3

**Summary:**

This paper introduces Terminal-Entropy Control (TEC), a novel approach to stochastic control that establishes a link between rate–distortion (RD) and entropic optimal transport (EOT) for continuous data sources under mean squared error (MSE) distortion. The authors demonstrate that the classical RD tradeoff between rate and distortion is equivalent to a tradeoff between control energy and the terminal-state differential entropy, and they characterize the optimal solution to TEC under certain regularity conditions. The authors also propose R2D2, a novel neural method for estimating the RD function and the reconstruction distributions using a simple diffusion model. Theory yields closed-form reconstruction distributions for mixtures; experiments validate on Gaussian, Gaussian mixtures, and small CIFAR-10 patches.

**Strengths:**

- Originality:  The paper introduces a novel dynamic formulation of the rate–distortion problem through the Terminal Entropy Control framework, which reinterprets RD optimization as a stochastic control process.
- Theoretical depth: The work provides substantial mathematical analysis and formal proofs, including clear optimality conditions via the Backward Heat Equation (BHE) and score-based drift characterization.
- Practical method (R2D2): Straightforward training; works from samples; empirically outperforms NERD/WGD on 1D Gaussian and matches analytic mixture results; scales to small image patches.

**Weaknesses:**

- Limited comparison: Experiments benchmark primarily against NERD and WGD only on the 1D Gaussian case. There is no comparison on gaussian mixtures or CIFAR-10, nor a discussion of why prior estimators cannot be adapted to these settings. Including why R2D2 generalizes better would enhance empirical credibility.
- Clarity and positioning:
    - Although the theoretical connection between RD, OT, and Schrödinger bridges is elegant, the manuscript could more explicitly contrast TEC with prior entropic OT formulations (e.g. WGD) and clarify where its generality or computational benefits concretely exceed them.
    - While the paper claims a “closed-form solution” for Gaussian mixtures, the actual rate–distortion values R(D) are still obtained via Monte Carlo or neural estimation. Clarifying this distinction would strengthen the paper’s theoretical claims.

**Questions:**

1. Relation to prior work (WGD / NERD):
Could the authors clarify why comparisons to WGD and NERD are only held for the 1D Gaussian case?
Is there a theoretical or computational reason these methods cannot handle Gaussian mixtures or CIFAR-10 patches?
2. Stability:
The backward heat equation underlying TEC is known to be ill-posed and potentially unstable. Could the authors elaborate on the numerical stability of training, especially for empirical datasets? In particular, how sensitive is R2D2 to the choice of entropy estimator (negentropy vs. KNIFE) in high-dimensional settings, and were any regularizations used to maintain stability? How strong or restrictive are the regularity constraints in practice—for instance, when $p_0$ represents real, high-dimensional data such as CIFAR-10 patches rather than smooth analytical densities?
3. Experimental setup:
Why are experiments performed on grayscale CIFAR-10 patches instead of full RGB images?

---

> ### Author Response · Authors · 2025-11-27
> **Response to Reviewer ZRmZ (1/2)**
>
> Thank you very much for your positive review and helpful comments.
>
> If our answer resolves your concerns, please consider to raise your score. We will be glad to address any remaining issue.
>
> Regarding your concerns:
>
> ---
>
> 1. **Experiments benchmark against baselines only on the 1D Gaussian case.**
>
> A: In our revision, we added such comparisons on real data (Fig. 5 in the revised manuscript). In the left pane we present the results on CIFAR dataset. In the right pane, we show results on additional `Speech' dataset, where we compare our method with baseline in the low-distortion regime.
>
> We may also note that there are no standard codes to run NERD nor WGD, hence conducting experiments with these baselines is difficult.
>
> ---
>
> 2. **Explicitly contrast TEC with  WGD. Why R2D2 generalizes better, what are computational benefits.**
>
> A:  Our answer here is 4-fold:
>
> **I)** First, compared to [Lei et al., 2022] and [Yang et al., 2024], our work suggests a fresh and fundamental view on RD theory as optimal control. This approach leads to new methods and analytical results, unknown for almost 70 years [Shannon, 1959] for information theorists.
>
> **II)** Our method R2D2 is based on modeling the controller function $u$ using a deep model. DNN architectures are flexible for increasing input dimensions. This also allows us to capture multiple positions on the RD-curve using a single network model, then traversing the tradeoff at evaluation phase without additional training.
> This is opposed to WGD for example, where working with high-dim data requires exponentially many more particles in the model, and choosing a different R-D point requires training a new model.
>
> **III)** Both WGD and NERD assume a logarithmic upper bound on the RATE values they can handle, where $R < \log M$ (for NERD, $M$ is the batch size, for WGD it is the number of atoms in the discrete probability approximation). This is a substantial  limitation since, in many practical cases,  low distortion  is required, which necesitates high rates (sometimes called the 'high-resolution regime'). Increaseing the rate in WGD and NERD requires exponentially larger batch size or atoms. Our method R2D2 does not suffer from this.
>
> **IV)** As we show in Fig.2, R2D2 outperforms other baselines on Gaussian data, where groundtruth $R(D)$ is known.
>
> We now highlight these points in the revision, (Remark 4.1, Sec. 5.2.2).
>
> ---
>
> 3. **The R(D) values  for Gaussian mixtures are obtained via estimation. Reconstruction distribution is in closed-form.**
>
> A: In the Introduction, we clearly state that "we find a closed-form solution **for the reconstruction distribution** of a Gaussian-mixture" (l.89). We will further clarify this point in the revision.
>
> Notwithstanding, we emphasize that no analytical results for the GMM setting were known to date, **at all**.
>
> ---
>
> 4. **Elaborate on the numerical stability of training, especially for empirical datasets. In particular, how sensitive is R2D2 to the choice of entropy estimator (negentropy vs. KNIFE) in high-dimensional settings.**
>
> A: Training R2D2 on high-dim data exhibited better stability when we used KNIFE rather than Negentropy method. This was mostly due to the sensitivity of the Negentropy term (Eq.(28)) to the numerical gradient operation (Torch grad). The KNIFE objective (Eq.(30)) was specially designed to allow such differentiations [Pichler et al. 2022].
>
> We note, however, that while the entropy estimator is an important component in Algorithm 1, the estimation itself is a technical step, orthogonal to our findings, and may be done in any chosen way. Any improvement in entropy estimation will also improve the RD estimation of R2D2.
>
> [Georg Pichler, Pierre Jean A Colombo, Malik Boudiaf, G\"unther Koliander, and Pablo Piantanida.
> A differential entropy estimator for training neural networks. In International Conference on
> Machine Learning, pages 17691–17715. PMLR, 2022]

---

> ### Author Response · Authors · 2025-11-27
> **Response to Reviewer ZRmZ (2/2)**
>
> 5. **Were any regularizations used to maintain stability?**
>
> A: No, we didn't take any such measures.
>
> ---
>
> 6. **How strong or restrictive are the regularity constraints when
> represents real, high-dimensional data?**
>
> A: As we discuss in (l.375), the assumptions of Thm. 3.2 are restrictive. Regularity conditions such as strict positive densities are common in the literature (see, e.g. [Chen, 2020], [Pavon, 1989]), as they allow the analysis of stochastic differential systems.
> In practice, however, real world datasets such as CIFAR-10 are not expected to possess such a smooth or positive density, since natural images are not dense in $R^n$ but rather concentrated around a low-dimensional manifold. Thus, we propose the R2D2 method, which only requires assumption A1, to address real world datasets.
>
> ---
>
> 7. **Why are experiments performed on grayscale?**
>
> A: Recall our experiments were focused on high rates (low distortions). As such, benchmarking on high-dimensional data required massive computational efforts, especially for running NERD and WGD baselines.
> Working with grayscale images allowed us to provide decent visual results while maintaining plausible running times and memory complexity, suitable to reasonable hardware capabilities.

---

### Author Response · Authors · 2025-11-27
**Revised manuscript and supplemental**

We first wish to express our gratitude to the reviewers for their valuable time and efforts.

Following your helpful comments, we now revised our manuscript. We hope that this answers all of your concerns, and will gladly address any other question or request.

---

Major modifications are (also marked in red in the revised text):

1. We better explain the motivation behind diffusion processes in RD theory, in the intro (l.74-77).
```
" It then becomes natural to investigate diffusion processes in the context of RD theory, as generative models casting the source probability to the distortion-optimal reconstruction distribution. As we show in this work, this fresh point of view reveals surprising analytical results as well as novel estimation methods. "
```

2. We discuss the motivation behind working in the low-distortion regime, right after Assumption A1 (l.237-239):
```
" Targeting the low-distortion regime is of
special interest because, in modern communication, high bandwidth channels are often available
(Chafii et al., 2023), and thus sources are compressed at high bitrates, allowing low distortion. "
```

3. We add Remark 4.1 regarding the ability of our Algorithm to work at high-rates. This highlights the advantages of our method compared to previous works.

4. We add an experiment on the "speech" dataset (Sec. 5.2.2)

5. We add comparisons to baselines on real-world datasets "Cifar" and "Speech" (Fig.5), with a special focus on the low-distortion regime, where we demonstrate how existing baseline are restricted compared to our method.

6. We add many of your suggested pointers and references to the intro and related work.

7. We attach code for reproducing Fig.2, arranged to run standalone as a single file, with the supplemental. We will similarly arrange the rest of our codes for our next revisions.

---

Thank you all again!

---

### Meta-Review · Area_Chair_v6iq · 2026-01-05

**Summary:**

Reviewer ZRmZ raises concerns about limited empirical comparison and clarity of positioning relative to prior entropic OT methods, as well as the claim of “closed-form” Gaussian-mixture results still relying on Monte Carlo estimation.

Reviewer GzUT questions the novelty relative to Dai Pra (1991) and suggests that parts of the theory and algorithm could be simplified using known results, also noting reproducibility concerns due to missing supplementary experiments.

Reviewer ndvv points to restricted scope (quadratic/low-distortion regime), scalability limits in higher dimensions, and a weak connection to broader machine-learning practice.

All reviewers except Reviewer GzUT acknowledge the strong theoretical contribution, solid mathematical grounding, and promising empirical performance of this work. I agree that the existence of Dai Pra (1991) does reduce the novelty of the present paper, particularly with respect to Theorem 3.2, to some extent. However, there still appear to be meaningful and distinguishable differences between the two approaches. Taken together, I am therefore inclined to recommend acceptance.

**Reviewer Concerns:**

The comments from Reviewers ZRmZ, GSoo, and ndvv appear to have been adequately addressed.

The rebuttal primarily focuses on Reviewer GzUT’s concern regarding the novelty of the present paper relative to Dai Pra (1991). It is unclear whether Reviewer GzUT has been fully convinced by the authors’ response. My sense is that the reviewer and the authors now largely agree on the technical aspects, but differ in their assessment of how much novelty the current work contributes beyond Dai Pra (1991).

**Reviewer Scores:**

Reviewer ZRmZ: 6
Reviewer GzUT: 2
Reviewer GSoo: 8
Reviewer ndvv: 8

All reviewers except Reviewer GzUT provided positive ratings. It is unclear how they will respond after considering Reviewer GzUT’s novelty concern regarding the present paper relative to Dai Pra (1991). My expectation is that the reviewers will largely maintain their original scores.

---

### Decision · Program_Chairs · 2026-01-26

Accept (Poster)